

# Multi-source global wetland maps combining surface water imagery and groundwater constraints

Ardalan Tootchi[1], Anne Jost[1], Agnès Ducharne[1]

[1] Sorbonne Université, CNRS, EPHE, Milieux environnementaux, transferts et interaction dans les hydrosystèmes et les sols, Metis, F-75005 Paris, France

*Correspondence to*: Ardalan Tootchi (ardalan.tootchifatidehi@upmc.fr)

**Abstract.** Many maps of open water and wetlands have been developed based on three main methods: (i) compiling national/regional wetland surveys; (ii) identifying inundated areas via satellite imagery; and (iii) delineating wetlands as shallow water table areas based on groundwater modelling. However, the resulting global wetland extents vary from 3% to 21% of the land surface area because of inconsistencies in wetland definitions and limitations in observation or modelling systems. To reconcile these differences, we propose composite wetland (CW) maps, combining two classes of wetlands: (1) regularly flooded wetlands (RFW) obtained by overlapping selected open-water and inundation datasets; and (2) groundwater-driven wetlands (GDW) derived from groundwater modelling (either direct or simplified using several variants of the topographic index). Wetlands are statically defined as areas with persistent near-saturated soil surfaces because of regular flooding or shallow groundwater. Seven CW maps were generated at the 15 arc-sec resolution (ca 500 m at the Equator) using geographic information system (GIS) tools and by combining one RFW and different GDW maps. To validate this approach, these CW maps were compared with existing wetland datasets at the global and regional scales. The spatial patterns were decently captured, but the wetland extents were difficult to assess against the dispersion of the validation datasets. Compared with the only regional dataset encompassing both GDWs and RFWs, over France, the CW maps performed well and better than all other considered global wetland datasets. Two CW maps, showing the best overall match with the available evaluation datasets, were eventually selected. These maps provided global wetland extents of 27.5 and 29 million km², *i.e.*, 21.1% and 21.6% of global land area, which are among the highest values in the literature and in line with recent estimates also recognizing the contribution of GDWs. This wetland class covers 15% of the global land area compared with 9.7% for RFW (with an overlap of ca. 3.4%), including wetlands under canopy/cloud cover, leading to high wetland densities in the tropics and small scattered wetlands that cover less than 5% of land but are highly important for hydrological and ecological functioning in temperate to arid areas. By distinguishing the RFWs and GDWs based globally on uniform principles, the proposed dataset might be useful for large-scale land surface modelling (hydrological, ecological and biogeochemical modelling) and environmental planning. The dataset consisting of the two selected CW maps and the contributing GDW and RFW maps is available from PANGAEA at https://doi.pangaea.de/10.1594/PANGAEA.892657



## 1 Introduction

Wetlands are valuable ecosystems with a key role in the carbon, water and energy cycles (Matthews and Fung, 1987; Richey et al., 2002; Repo et al., 2007; Ringeval et al., 2012). Water retention in wetlands leads to lower and delayed runoff peaks, higher base flows and evapotranspiration, which directly influence climate (Bierkens and van den Hurk, 2007; Lin et

al., 2016). Wetlands also serve to purify pollutions from natural and human sources, thus maintaining clean and sustainable water for ecosystems (Billen and Garnier, 1999; Dhote and Dixit, 2009; Curie et al., 2011; Passy et al., 2012). Despite their widely recognized importance, no consensus exists on wetland definitions and their respective areal extents among the reviewed literature (Table 1). Based on tens of definitions, the extents range from regions with relatively shallow water tables (National Research Council, 1995; Kutcher, 2008; Ramsar, 2009) to areas with permanent inundation such as lakes (lacustrine

wetlands) with depths of several metres. The reasons for this ambiguity are a diversity of scientific points of views as well as the complexity of classification in transitional land features and temporally varying land features under human influences (Mialon et al., 2005; Papa et al., 2010; Ringeval et al., 2011; Sterling et al., 2013; Hu et al., 2017; Mizuochi et al., 2017).

The first global wetland maps were developed based on compilation of regional archives and estimates. Matthews and Fung (1987) developed a 1° resolution wetland map based on vegetation, soil properties and inundation fractions that covered

ca. 4% of the land. Finlayson et al. (1999) based their estimates on surveys and the Ramsar global inventory in which wetlands cover 9.7% of the land area. Later, the global lakes and wetlands database (GLWD) was developed at 30 arc-sec resolution (~1 km at the Equator) by compiling several national and regional wetland maps with a global cover of 6.9% of land area, excluding Antarctica and glaciated lands (Lehner and Döll 2004). Because satellite imagery permits homogeneous observation of land characteristics, this method has been favoured for mapping of water-related features in recent decades. Satellite imagery

at visible wavelengths reports that 1.6 to 2.3% of Earth's land is permanently under water (Verpoorter et al., 2014; Feng et al., 2015; Yamazaki, et al., 2015; Pekel et al., 2016) but with large disagreements (Nakaegawa, 2012), and inundations under densely vegetated and clouded areas are often missed (Lang and McCarty 2009). Longer wavelengths in the microwave band (*e.g.,* L and C bands) penetrate better through the cloud and vegetation layer and supply dynamic observations of inundated zones, usually with a trade-off between high resolution with a low revisit rate or domain extent (Li and Chen, 2005; Hess et

al., 2015) and coarse resolution with a high revisit rate up to global coverage (Prigent et al., 2007; Papa et al., 2010; Schroeder et al., 2015; Parrens et al., 2017). Recent progress has been achieved by downscaling or correcting the latter products using higher-resolution information. Fluet-Chouinard et al. (2015) developed the global inundation product GIEMS-D15 by downscaling the 0.25° multi-satellite wetland fractions of Prigent et al. (2007) using 15 arc-sec topography, with a global long-term maximum inundation fraction of 13%. Poulter et al. (2017) corrected the wetland fractions of the surface water microwave

product series (SWAMPS: Schroeder et al., 2015) by merging them with the those obtained at 30 arc-sec from GLWD.

However, regardless of the wavelengths, wetlands derived from satellite imagery almost always represent inundated areas and overlook other types of wetlands where soil moisture is high but the surface is not inundated (Maxwell and Kollet, 2008; Lo and Famiglietti, 2011; Wang et al., 2018). The method most frequently used to delineate these wetlands is water table depth (WTD) modelling. Direct groundwater (GW) modelling (*e.g.,* Miguez-Macho and Fan 2012) requires in-depth knowledge of

the physics of water movement, topography at a sufficiently high resolution, climate variables, subsurface characteristics and observational constraints (Fan et al., 2013; de Graaf et al., 2015). Simplified GW models based on the topographic index (TI) of TOPMODEL (Beven and Kirkby 1979) require less extensive input, and they have also been used to map wetlands (*e.g.*, Gedney and Cox, 2003). Using the topography, TI can be calculated as follows:

$$TI = ln\left(\frac{a}{tan(\beta)}\right), \tag{1}$$

where $a$ $(m)$ is the drainage area per unit contour length, and $tan(\beta)$ is the local slope at the desired pixel. The TI index is often presented as a wetness index (Wolock and McCabe, 1995; Sørensen et al., 2006) because high values are found over flat regions with large drainage areas corresponding to a high propensity for saturation. Other environmental characteristics



such as climate and soil or underground properties can also be used in the TI formulation to detect wetlands in areas where topography is not the primary driver of the water budget, such as wetlands in uplands and over clayey soils or thin active layers in the permafrost region (*e.g.,* Saulnier et al., 1997; Mérot et al., 2003; Hu et al., 2017).

A major challenge in identification of wetlands through GW modelling is the definition of thresholds on TI or WTD for separation of wetland from non-wetland areas. The thresholds are often calibrated to reproduce the extent of documented wetlands in a certain region and are subsequently extrapolated for larger domains. This strategy was proven successful at the basin scale (*e.g.*, Curie et al., 2007), but it has been shown to be ineffective at larger scales because it is not possible to uniquely link TI values to soil saturation levels across different landforms and climates (Marthews et al., 2015). Hu et al. (2017) produced a global wetland map by calibrating TI thresholds for every large basin of the world based on land cover maps, as

pioneered over France due to independent TI threshold calibration in 22 hydro-ecoregions using soil type datasets (Berthier et al., 2014). Uniform WTD thresholds (0 cm for inundated areas and 25 cm for wetlands) are applied in the only example (to our knowledge) of direct global GW modelling for wetland delineation (Fan and Miguez-Macho, 2011 and Fan et al., 2013). All these datasets based on GW modelling estimate the wetland fraction as much higher than those based on inventories and satellite imagery (Hu et al., 2017: 22.6%, Fan et al., 2013: 15% of the land surface area). It must be emphasized that adjustment

of wetland thresholds, both for directly modelled WTD and TI, always implies subjective choices and can result in over/underestimation of wetland extents or unrealistic wetland distribution patterns.

The scientific objective of the current work is to develop a comprehensive global wetland dataset based on a unique and applicable wetland definition for use in hydrological and land surface modelling. Based on the above analysis, our rationale is that inundated and groundwater-driven wetlands must both be considered to realistically capture the wetland patterns and

extents. This approach leads to a definition of wetlands as areas that are persistently saturated or near saturated because they are regularly subject to inundation or shallow water tables. Although these two classes partially overlap and share similar environmental properties, they cannot be detected using a single method. Thus, we rely on data fusion methods, which have proven advantageous in developing high-quality products by merging properties from various datasets (Fritz and See, 2005; Jung et al., 2006; Schepaschenko et al., 2011; Pérez-Hoyos et al., 2012; Tuanmu and Jetz, 2014), including wetland mapping

(Ozesmi & Bauer, 2002; Friedl et al., 2010; Poulter et al., 2017). In this framework, we tested several composite wetland (CW) maps, all constructed at 15 arc-sec resolution, by merging two complementary classes of wetlands: (1) regularly flooded wetlands (RFWs), where surface water can be detected at least once a year through satellite imagery; and (2) groundwater-driven wetlands (GDWs) based on groundwater modelling.

The main assumptions underlying the composite wetland maps are detailed in Sect. 2, together with the involved datasets.

Subsequently, Sect. 3 sequentially presents the construction of the RFW, GDW, and CW maps, with preliminary analyses of their features and uncertainties. In Sect. 4, we compare the CW maps with several validation wetland datasets, globally and in several areas with contrasting climates and wetland fractions, to show that the combination of RFWs and GDWs a consistent wetland description throughout the globe. This comparison allows us to select two CW maps with better overall performances, used to discuss the role of GDW in Sect. 5. Finally, the availability and potential applications of the composite maps are

presented in Sect. 6, while Sect. 7 summarizes the advantages and limitations of the approach and gives perspectives on future developments.

## 2 Methods and data

### 2.1 Wetland definition and general mapping strategy

The wetland definition behind the composite maps is focused on hydrological functioning, and we aim to include both

seasonal and permanent wetlands as well as shallow surface water bodies (including rivers, both permanent and intermittent).





Surface water bodies and wetlands are often hydrologically connected, and the transition between them is not sharp and varies seasonally. Moreover, these features are difficult to separate based on observations (either in situ or remote), and no dedicated exhaustive dataset is currently available (Raymond et al 2013; Schneider et al., 2017). Inclusion of the shallow surface water bodies (in the RFW map) is compatible with the Ramsar classification, but we depart from this approach with respect to large

permanent lakes, which are excluded from all input datasets to RFW and GDW maps (Sect. 2.2) because of their distinct hydrology and ecology compared with wetlands. In contrast, groundwater-driven wetlands can remain wet without inundation due to the presence of shallow water tables. As further discussed in Sect. 3.2, these areas are defined in this study as areas where the mean annual WTD is less than 20 cm, following similar assumptions in the literature (U.S. Army Corps of Engineers, 1987; Constance et al., 2007; Tamea et al., 2010; Fan and Miguez-Macho, 2011).

Another feature of the proposed wetland maps is that they are static. As stated in Prigent et al. (2007), the maps represent the "climatological maximum extent of active wetlands and inundation" (for CWs and RFWs, respectively), *i.e.,* the areas that happen to be saturated or near saturated sufficiently frequently to develop specific features of wetlands (high soil moisture over a significant period of the year leading to reducing conditions in selected horizons and specific flora and fauna).

Based on the above definitions and assumptions, we use GIS tools to construct several composite wetland maps as the

overlap (union) of the following:

- One RFW map developed by overlapping three surface water and inundation datasets derived from satellite imagery in an attempt to fill the observation gaps (Sect. 3.1);
- One GDW map out of seven, all derived from GW modelling (either direct or simplified based on several TI versions) and meant to sample the uncertainty of the GDW contribution (Sect. 3.2).

In this process, many layers were developed and are summarized in Table 2 and detailed in Sect. 3. Input datasets to RFWs and GDWs are presented in Sect. 2.3 and 2.4 respectively, and several independent validation datasets, global and regional, are presented in Sect. 2.5. It must be noted that all involved layers were resampled at 15 arc-sec resolution, which is within the resolution range of state-of-the-art wetland-related datasets. In doing so, we used an "all-or-nothing" approach, *i.e.,* the pixels are either fully recognized as wetland (or lake) or not at all, based on the dominant type if the input data is finer than 15 arc-

sec (Sect. 2.6). In the remainder of this paper, the wetland percentages of the land surface area always exclude lakes, the Caspian Sea, the Greenland ice sheet and Antarctica (unless otherwise mentioned). For this reason, these percentages and areas might be different from those shown in Table 1, which are indicated in each original paper or data description.

### 2.2 Lakes

To distinguish large permanent lakes from wetlands, we used the HydroLAKES database (Messager et al., 2016), which

was developed by compiling national, regional and global datasets (Fig. 1a). This database consists of more than 1.4 million individual polygons for lakes with a surface area of at least 10 ha, covering 1.8% of the land surface area. This value is smaller than those in other recent databases that account for smaller water bodies: 2.5% in G3WBM (Yamazaki et al., 2015) for water bodies above 0.8 ha and 3.5% in GLOWABO (Verpoorter et al., 2014) for those above 0.2 ha. These two datasets do not differentiate lakes from other surface water elements and using them as a mask would lead to exclusion of shallow inundated

portions of wetlands (*e.g.,* Indonesian mangroves or Ganges floodplains). It must also be noted that the small water bodies tend to be overlooked after dominant resampling to 15 arc-sec resolution (Sect 2.6), unless they are sufficiently numerous in a pixel. Therefore, the lake mask covers 1.7% of the land area compared with 1.8% in the original HydroLAKES database. This map also shows that most of the lakes are located in the northern boreal zones (more than 60% of lakes area are located north of 50°N), in agreement with the other lake databases.





### 2.3 Input to RFW map: Inundation datasets

#### 2.3.1 ESA-CCI land cover

This dataset succeeds the GlobCover dataset based on the data from the MERIS sensor (onboard ENVISAT) collected at high resolution for surface water detection, together with the SPOT-VEGETATION time series (Herold et al., 2015) to aid in
distinguishing wetlands from other vegetation covers. Global land cover maps at approximately 300 m (10 arc-sec) resolution deliver data for three 5-year periods (1998-2002, 2003-2007 and 2008-2012). The extents of water bodies slightly changed between the first 5-year period and the third one (such as shrinking of the Aral Sea area by more than 55%), but the extent of wetland classes (permanent wetlands and flooded vegetation classes) did not change significantly (the variation in wetland classes throughout these periods is less than 3% of the total wetlands area). We acquired the last epoch data to represent the
current state of wetlands (Fig. 1b). In ESA-CCI, wetlands are mixed classes of flooded areas with tree covers, shrubs or herbaceous covers plus inland water bodies, covering 3% of the Earth land surface overall.

#### 2.3.2 GIEMS-D15 (Fluet-Chouinard et al., 2015)

Prigent et al. (2007) used multi-sensor satellite data, including passive and active microwave measurements, together with visible and near-infrared reflectance to map the monthly mean inundated fractions at 0.25° resolution for a 12-year period
(1993 to 2004). This dataset (GIEMS) gives the minimum and maximum extent of the inundated area (including wetlands, rivers, small lakes, and irrigated rice). Fluet-Chouinard et al. (2015) used the GLC2000 land cover map (Bartholomé and Belward 2005) to train a downscaling model for GIEMS at 15 arc-sec resolution based on the HydroSHEDS digital elevation model (Lehner et al., 2008) and developed three static datasets for mean annual minimum, mean annual maximum and long-term maximum extent of the inundated areas (covering 3.9%, 7.7% and 10.3% of the land surface area, respectively). In this
study, we assumed that the mean annual maximum extent was the best representative measure for wetlands. In the following, GIEMS-D15 always indicates the mean annual maximum of GIEMS-D15 (Fig. 1c). Higher-resolution (3 arc-sec) downscaling of GIEMS has been recently developed (Aires et al., 2017), but we overlooked this source because we focused our study on the 15 arc-sec resolution.

#### 2.3.3 JRC surface water (Pekel et al., 2016)

The JRC surface water products are a set of high-resolution maps (1 arc-sec ~ 30 m) for permanent water and also for seasonal and ephemeral water bodies. These products are based on analysis of Landsat satellite images (Wulder et al., 2016) over a period of 32 years (1984-2015). Each pixel was classified as open water, land or non-valid observation. Open water is defined as any pixel with standing water, including fresh and saltwater. The study also quantifies the conversions, mostly referring to changes in state (lost or gained water extents, conversions from seasonal to permanent, etc.) during the observation
period. In this study, we used the maximum surface water extent, which consists of all pixels that were under water at least once during the entire period, covering almost 1.5% of the Earth land surface area (Fig. 1d).

### 2.4 Input to GDW maps

#### 2.4.1 Water table depth estimates (Fan et al., 2013)

Fan et al. (2013) performed global GW modelling to estimate the water table depth at 1 km resolution. This model assumes
a steady flow, and lateral water fluxes are calculated using the Darcy's law and the Dupuit-Forchheimer approximation for 2-D flow. Elevation is described at 30 arc-sec resolution (by HydroSHEDS south of 60° N and otherwise by ASTER/NASA-JPL), and the recharge rates were modelled using the WaterGAP model (Döll and Fiedler, 2008) based on contemporary





meteorological forcing (1979-2007). To estimate subsurface transmissivity, the soil hydraulic conductivities were derived from the global Food and Agriculture Organization (FAO) digital soil maps (5 arc-min resolution) and US Department of Agriculture (USDA) soil maps over the United States (30 arc-sec resolution) and subsequently assumed to decay exponentially with depth from the thin soil layer (2 m) down as a function of the local topographic slope. The decay factor is also adjusted

for the permafrost region using an additional thermic factor (smaller transmissivity in permafrost areas). The modelled WTD was constrained to observations available to the authors (more than one million observations with 80% of them located in North America). The resulting dataset suggests vast areas with a shallow water table over the tropics, along the coastal zones, and in boreal areas of North America and Asia (almost 15% of the land area for WTD ≤ 20 cm).

**2.4.2 Global TI (Marthews et al., 2015)**

Marthews et al. (2015) produced a global map of TI at 15 arc-sec resolution using the original formulation in Beven and Kirkby (1979), as in Eq (1), and used two global high-resolution digital elevation models (DEMs), *viz.* HydroSHEDS (Lehner et al., 2008) and Hydro1k (U.S. Geological Survey, 2000) at 15 and 30 arc-sec resolution, respectively. Hydro1k is used to fill the lack of information in HydroSHEDS north of 60°N, which is outside of the SRTM (Shuttle Radar Topography Mission) coverage. Because index values depend on pixel size, which varies with latitude, those researchers also applied the

dimensionless topographic wetness index correction of Ducharne (2009) to transform the index values to equivalents for a 1-meter DEM.

**2.4.3 CRU climate variables**

To assess the impact of climate on wetlands, we used the Climatic Research Unit (CRU) monthly meteorological datasets. These datasets cover all land area from the beginning of the twentieth century (Harris et al., 2014). CRU climate time series

are gridded to 0.5° resolution based on more than 4000 individual weather station records. To include a climate factor in the TI formulations, the time series of selected climate variables (*i.e.*, precipitation and potential evapotranspiration based on the Penman-Monteith equation) are extracted for the contemporary period (1980-2016).

**2.4.4 GLHYMPS (Gleeson et al., 2014)**

GLHYMPS is a global permeability and porosity map based on high-resolution lithology (Hartmann and Moosdorf, 2012).

The permeability dataset and its derived hydraulic conductivity ($K_s$) estimates are given in vector format with an average polygon size of approximately 100 km². As noted by the developers of GLHYMPS (Gleeson et al., 2011, 2014), "lithology maps represent the shallow subsurface (on the order of 100 m)", and thus hydraulic conductivity estimates are valid for the first 100 m of the subsurface layer. Thus, we estimated transmissivity as the integral of this constant $K_s$ over these 100 m and used it to check whether use of the available transmissivity datasets in TI formulations can improve global wetland

identification. It should be noted that the hydraulic conductivity dataset has two versions: with and without the permafrost effect. To consider the permafrost effect, Gleeson et al. (2014) used maps of the permafrost zonation index (PZI) from Gruber (2012) and homogenously assigned a rather low hydraulic conductivity ($K_s = 10^{-13} \, m/s$) for areas with PZI > 0.99. For our calculations, we rasterized the vector polygons of 🮲🯰 without the permafrost effect to 15 arc-sec resolution.

**2.5 Validation datasets**

Two global and two regional wetland datasets were used to assess the validity of the CW maps, and none of them was used as inputs to the composite wetland maps to ensure an independent evaluation of the strengths and weaknesses of the CW maps.



### 2.5.1 GLWD-3 (Lehner and Döll, 2004)

GLWD is a global lakes and wetlands dataset based on aggregation of regional and global land cover and wetland maps. This dataset contains three levels of information, and the most inclusive one is GLWD-3, which is in raster format. This dataset has an original 30 arc-sec resolution and contains 12 classes for lakes and wetlands (maps and details are given in the supplementary information, Sect. S1 and Fig. S1). For large zones prone to water accumulation but without solid information on existing wetlands, fractional wetland classes are defined (together they cover 4% of the land surface area). This is particularly the case within the Prairie Pothole Region in North America and the Tibetan plateau in Asia. Depending on the interpretation of fractional wetlands (by taking either the minimum, mean or maximum fraction of the ranges), wetlands cover between 5.8 and 7.2% of the land surface area. In this paper, we take the mean fraction in these areas, leading to a total wetland extent of 6.3% of the land surface area.

### 2.5.2 Global wetland potential distribution (Hu et al., 2017)

Hu et al. (2017) proposed a potential wetland distribution using a "precipitation topographic wetness index" based on a new TI formulation in which the drainage area is multiplied by the mean annual precipitation. This formulation is based on the concept of the topography-climate wetness index (Mérot et al., 2003) in which the effective precipitation was introduced as the climate factor. The new index is calculated at 1 km resolution using GTOPO30 elevation data developed by the USGS. Wetlands are categorized into "water" and "non-water wetlands" based on regionally calibrated thresholds for each large basin of the world (level-1 drainage area of Hydro1k) using a sample trained adjustment model. The water classes of several land cover datasets are used to train the model for the "water" threshold, and the model for the "non-water wetland" threshold is trained on the regularly flooded tree cover and herbaceous cover categories (additional details are available in the supplementary information, Sect. S1 and Fig. S2). The global coverage of the "water" and "non-water wetland" classes in Hu et al. (2017) is 22.6% of the Earth land surface area (excluding lakes, Antarctica and the Greenland ice sheet), considering no loss due to human influence. This dataset gives the largest wetland extent within the accessible literature, with notably large water wetlands in South America and large non-water wetlands in Central Asia and Northern American continent. In this paper, we used the union of the "water" and "non-water wetlands" classes of this dataset for further evaluations.

### 2.5.3 Amazon basin wetland map (Hess et al., 2015)

Hess et al. (2015) used the L-band synthetic aperture radar (SAR) data from the Japanese Earth resources satellite (JRES-1) imagery scenes at a 100 m resolution to map wetlands during the period 1995-1996 for high and low water seasons. The studied domain excludes zones with altitudes higher than 500 m and corresponds to a large fraction of the Amazon basin (87%). Wetlands are defined as the sum of lakes and rivers (both covering 1% of the basin area) and other flooded areas plus zones not flooded but adjacent to flooded areas and sharing wetland geomorphology. The flooded fraction of wetlands varies considerably (from 38% to 75%.) between the low and high-water season. The total maximum mapped wetland area extends over 0.8 million km$^2$ and is used in evaluation of CW maps in this study.

### 2.5.4 Modelled potentially wet zones of France

The map of potentially wet zones in France (les Milieux Potentiellement Humides de France Modélisée: MPHFM; Berthier et al., 2014) constructed at 50 m resolution is based on the topo-climatic wetness index (Mérot et al., 2003) and the elevation difference to streams using the national high resolution DEMs. Meteorological data for calculation of the topo-climatic index (precipitation and potential evaporation rates; see further details in Sect. 3.2.2) are taken from the SAFRAN atmospheric reanalysis (Vidal et al., 2010) at 8 km resolution. Index thresholding for wetland delineation is performed



independently in 22 hydro-ecoregion units and delimited based on lithology, drainage density, elevation, slope, precipitation rate and temperature. The wet fraction defining the threshold in each hydro-ecoregion is the fraction of hydromorphic soils taken from national soil maps at 1:250,000 (InfoSol, 2013). Additionally, the elevation difference between land pixels and natural streams was used to separate large streambeds and plain zones, which are difficult to model with indices based on topography. Based on MPHFM potential, wetlands extend over almost 130,000 km$^2$ of France (23% of the area of metropolitan France). The dataset was validated against available pedological point data (based on profiles or surveys) available over France. These point data are classified into wetlands and non-wetlands for the validation procedure. This procedure used statistical criteria such as spatial coincidence (number of correctly diagnosed points over total number of points) and Kappa coefficient (modelling error compared with a random classification error).

**2.6 Data processing**

To project, resample, intersect/overlap and convert different datasets used in wetland mapping in this study, we relied on ArcMap software (Esri, ArcGIS Desktop: Release 10.3.1 Redlands, CA) and its different tools. All datasets were projected to a WGS84 equi-rectangular coordination system and subsequently resampled to a single resolution for facilitated fusion and comparison. The resulting raster datasets were processed with ArcMap tools available in almost any GIS software such as QGIS (Table 3).

**2.6.1 Input resampling to 15 arc-sec**

The final resolution of the maps is targeted to 15 arc-sec (~500 m at the Equator) for consistency with the available water datasets. Therefore, all datasets were resampled to 15 arc-sec resolution. For datasets at coarser resolutions, each coarse pixel is disaggregated to 15 arc-sec while retaining the same value. For datasets with higher resolution than the target resolution (ESA-CCI land cover and JRC surface water), the resampling (aggregation) process was based on a majority area function such that the resampled 15 arc-sec pixel has the value of the dominant feature (covering the largest fraction).

**2.6.2 Results aggregation to coarser resolution**

Each 15 arc-sec global raster contains more than 80,000 pixels along a circle of 360° of longitude, and wetlands can exhibit notably small-scale patterns (*e.g.*, patchy or river-like). To facilitate visual inspection, we calculated the mean wetland densities at 3 arc-min grids for most of the maps presented in this work. The same 3 arc-min resolution (~6 km at the Equator) was used in calculating the spatial correlations. For zonal wetland area distributions, the area covered by wetlands in each 1° latitude band is displayed.

**3 Construction of composite wetland maps**

**3.1 Regularly flooded wetland (RFW) maps**

**3.1.1 Mapping by data fusion**

To identify the RFWs, we overlapped carefully selected datasets of surface water, land cover and wetlands, namely, the ESA-CCI land cover, GIEMS-D15 inundation surface, and the maximum water extent in JRC surface water. These datasets were selected to include different types of data acquisition. The idea behind the fusion approach chosen in this work is that wetlands identified by the different datasets are all valid despite their uncertainties, although none of them are exhaustive. As a result, use of multiple inundation datasets fills the observational gap. Several other surface water datasets exist that were not





used in this work, either because they mostly consist of lakes or because they rely on similar methodologies (Verpoorter et al., 2014; Yamazaki et al., 2015).

### 3.1.2 Geographic analysis

Overall, the RFW map covers 9.7% of the land surface area (12.9 million km$^2$) including river channels, deltas, coastal wetlands and flooded lake margins (Fig. 1e). Areal coverage of the RFWs is by definition larger than the area of wetlands in all three selected datasets (Fig. 1b-d). The contribution of each input inundation datasets to the RFW map is fairly different. The shared fraction of the three composing elements is minuscule (5% of the total RFW land surface area coverage), showing the vast disagreement among them. As such, 58% of RFWs are solely sourced from GIEMS-D15, consisting mostly of South-east Asian floodplains, North-east Indian wet plains and rice paddies and wetlands in the Prairie Pothole Region (in Northern US and Canada). However, the ESA-CCI contribution is found mainly over the Ob River basin. Due to its high resolution, JRC surface water adds small-scale wetlands such as patchy wetlands, small ponds and oases (0.4% of the land surface area).

In terms of zonal distribution, 31% of the RFWs are concentrated north of 50°N with most of the wetlands formed in the Prairie Pothole Region and Siberian lowlands. Cold and humid climates and the poorly drained soils of the boreal forest regions in Northern Canada on the Precambrian shield are the main hotspots of peat in the American continent. The same situation exists in the western Siberian plains as well. The second zonal peak in RFWs lies between 20°N and 33°N, where the major contributors are the vast floodplains surrounding the Mississippi, Brahmaputra, Ganges, Yangtze, and Yellow Rivers and Mesopotamian marshes. A total of 30% of the world's RFWs are found in tropical regions (20°N to 20°S), concentrated mainly in the Amazon, Orinoco and Congo River floodplains and in inundated portions of wetlands such as the Sudd swamp in South Sudan.

## 3.2 Groundwater-driven wetland (GDW) maps

### 3.2.1 Mapping based on WTD

Due to a lack of integrated, standardized and globally distributed WTD observations, a sound approach to location of groundwater-driven wetlands is the use of available global direct GW modelling results. In this study, we used the global WTD estimations of Fan et al. (2013), and the resulting wetland map is denoted as GDW-WTD. We assumed the mean annual WTD in wetlands to be less than 20 cm, following similar assumptions in the literature (U.S. Army Corps of Engineers, 1987; Constance et al., 2007; Tamea et al., 2010; Fan and Miguez-Macho, 2011). This results in a wetland area extending over 15% of the land surface, with large wetlands in the northern areas and the Amazon basin (Fig. 2a). We also performed a sensitivity analysis on the areal fraction of wetlands with different WTD thresholds (supplementary section S2, Fig. S3 and S4), revealing that the variation in total wetland fraction is quite weak (between 13.7% and 16.7%) for thresholds ranging from 0 to 40 cm. Therefore, a 20 cm threshold appears to be a credible representative value. However, the wetland fraction rapidly increases for deeper thresholds, showing that a clear distinction exists between shallow WTD areas (wetlands according to our definition) and the remainder of the land.

### 3.2.2 Mapping based on various TIs

#### Three wetness index formulations

Topography is often not sufficient for wetland identification because climate and subsurface characteristics also control water availability and vertical drainage. Using the original TI formulation in Eq (1), high index zones might coincide with flat arid areas, or inversely, low index values might occur at wetland zones with small upstream drainage areas over a shallow impervious layer. Several studies have focused on improving the topographic wetness index for wetland delineation by



including other environmental factors or modifying the formulation of the wetness index (Rodhe and Seibert, 1999; Mérot et al., 2003; Manfreda et al., 2011). In this study, we used three variants of TI. The global dataset of TI developed by Marthews et al. (2015) at 15 arc-sec resolution is used to supply the original TI and as a base map to derive two other variants of the index.

The first variant index is the TCI (topography-climate wetness index, inspired by Mérot et al., 2003):

$$TCI \quad = ln\left(\frac{a \cdot P_e}{tan(\beta)}\right) \quad = TI \; + \; ln(P_e), \tag{2}$$

where $P_e$ is the mean annual effective precipitation (in metres). The effective precipitation is first defined at the monthly time step as the monthly precipitation $P_{m,y}$ (meters) for month $m$ and year $y$ that is not evaporated or transpired using the monthly potential evapotranspiration $EP_{m,y}$ (meters) as a proxy for total evapotranspiration:

$$P_{m,y}^e = max(0, P_{m,y} - EP_{m,y}). \tag{3}$$

$P_e$ is subsequently calculated as the sum of the 12 pluri-annual means of monthly effective precipitation. The required climatic variables are taken from the CRU monthly meteorological datasets (Sect. 2.2.3) for 1980-2016 to represent the contemporary period.

The second variant index (known as TCTrI for topography-climate-transmissivity index) is constructed by combining the

effect of heterogeneous transmissivity (Rodhe and Seibert, 1999) with the above TCI:

$$TCTrI = ln\left(\frac{a \cdot P_e}{Tr.tan(\beta)}\right) = TI \; + \; ln(P_e) - ln(Tr), \tag{4}$$

where $Tr$ (m²/s) is the transmissivity calculated by vertically integrating a constant $K_s$ (saturated hydraulic conductivity in m/s) from GLHYMPS over the first 100 m below the Earth's surface (Sect. 2.4.4).

**Two index thresholds for two global GDW fractions**

Similar to many studies (Rodhe and Seibert 1999; Curie et al., 2007; Hu et al., 2017), we define TI-based wetlands as the pixels with TI above a certain threshold, defined to match a certain fraction of total land. In doing so, we prescribe the global GDW fraction as a chosen value, and the various TI formulations only change the geographic distribution of the corresponding wetlands. To apprehend the uncertainty related to the choice of the global GDW fraction, we tested two choices within the bounds derived from the global WTDs of Fan et al. (2013). In the first approach, we set the TI threshold such that the wet

pixels (with high index values) cover 15% of the land surface area, such as the fraction of WTD ≤ 20 cm according to Fan et al. (2013). The corresponding maps are noted as GDW-TI(15%), GDW-TCI(15%) and GDW-TCTrI(15%) in Table 2 and show fairly different patterns (Fig. 2b-d). The second approach assumes that the total wetland extent (this time including both GDWs and RFWs) covers 15%. The TI thresholds are subsequently set such that the union of RFWs and GDW-TI (TCI/TCTrI), *i.e.,* the composite wetlands, has the same extent as GDW-WTD. The resulting GDWs cover between 6 and

6.6% of the land area depending on the TI formulation and level of overlap with RFWs (Table 4) and are noted as GDW-TI(6%), GDW-TCI(6.6%), and GDW-TCTrI(6%). The patterns of these three maps are highly similar to those of GDW-TI(15%), GDW-TCI(15%) and GDW-TCTrI(15%) with diminished extents and densities (Fig. 2e-g).

### 3.2.3 Comparison of the proposed GDW maps

As shown in Table 2, seven GDW maps are developed, consisting of GDW-WTD (Sect. 3.2.1) and six GDW-TIs (Sect.

3.2.2). The GDW-WTD map contains high wetland extents over the northern latitudes (Fig. 2a), in contrast to the other six GDW maps. The diagnosed wetlands of GDW-TI maps (Fig. 2b, e) are equally distributed over well-known arid areas such as the Sahara and Kalahari Desert, Australian shield and Arabian Peninsula as in wet regions such as West Siberian plain and Northern Canada (Fig. 2b, e). As a result, for a given threshold (15% in Fig. 3a), the distribution of wetlands derived from the simple TI is nearly uniform over different latitudes. Lower thresholds on TI variants (Fig. 2e-g and Fig. 3b) obviously result





in a smaller wetland extent with no major change in the zonal pattern when the wet fraction threshold changes from 15% to 6% (Fig. 2b-d and Fig. 3a, b).

Introducing a climate factor in the form of effective precipitation in GDW-TCI(6.6%) and GDW-TCI(15%) increases the value of the wetness index in wet areas and decreases it in dry climates (Fig. 2c, f and Fig. 3a, b). Therefore, previously
diagnosed wetlands with TI in dry climates disappear and transfer to regions with wet climates (such as the Amazon basin and South Asia). However, because transmissivity values sharply change by several orders of magnitude over regions with small permeability, the patterns of GDW-TCTrI maps are nearly replicas of the low hydraulic conductivity distribution in GLHYMPS (*e.g.*, large diagnosed wetlands in North America and central Asia; Fig. 2d, g). Although at times GDW-TCTrI coincides with famous wetlands such as the Pampas in South America (Fig. 2d, g and near 25°S in Fig. 3a), diagnosed wetlands
extend far beyond the actual wet regions into neighbouring arid/semi-arid zones, *e.g.,* vast diagnosed wetlands in the western Siberian lowlands extend southward towards the Kazakh upland arid zones. In the absence of precise and consistent subsurface characteristics information (particularly for cold areas), GDW-TCTrI shows low wetland densities in zones with the known effect of transmissivity, such as the Hudson Bay lowlands and the Prairie Pothole Region.

### 3.3 Composite wetland (CW) maps

Each GDW map was overlapped with the RFW map to generate seven CW maps. Equi-resolution raster pixels of RFWs and GDWs were aligned to coincide exactly with each other. The resulting composite wetland maps are named with respect to their contributing GDW component (Table 2), *e.g.*, the composite map containing RFW and GDW-TI(6%) is known as CW-TI(6%). These composite wetlands cover between 15% and 22% of the land surface area. Each CW map contains RFWs and GDW and thus wetlands shared by both wetland classes (the intersection). The intersection between GDW and RFW maps is
larger for TCI-based maps and GDW-WTD (almost one third of RFWs intersect with these GDW maps) compared with TI and TCTrI-derived GDW maps (Table 4). These intersection zones are further discussed in Sect. 4. The wetland extent in CWs is by definition larger than both RFW and GDWs, and their spatial patterns depend on the contribution percentage of each component. As an example, in CW-TCI(15%), over most latitudes, the spatial pattern is similar to that of RFW, except over the tropical zones where GDWs are far more extensive than RFWs, thus shaping the general latitudinal pattern (Fig. 3c).
Changing the percentage of GDWs (between 6 and 15%) based on different TI formulations increases the wetland fraction of the CW maps to between 5.3% and 12.5% of the land area, but it does not considerably change their overall latitudinal pattern (Fig. 3d, e). In RFW, large wetlands are present between 25°N and 35°N (Fig. 3c), whereas in all GDW maps, the wetland extents over these latitudes are smaller than in other wetland regions (Fig. 3a, b).

### 4 Validation

**4.1 Spatial similarity assessment**

A difficulty inherent in the validation of any wetland map is the vast disagreements among available datasets and estimates. In this paper, we used independent validation datasets (explained in Sect. 2.5) that are not used in any step as input to our final products, but we made an exception for the GDW-WTD (derived from Fan et al., 2013), which is an input to CW-WTD but is independent from the RFW and other CW maps. All seven developed CW maps and the RFW map were evaluated
using the spatial coincidence, Jaccard index and spatial Pearson correlation coefficient with respect to the validation datasets over the globe and in several regions, the latter of which are discussed in detail below.

The first evaluation criterion of spatial coincidence (SC) is defined as the fraction of pixels identified as wet in a validation dataset that are also detected in the composite wetland dataset:

$$SC = \frac{Area\ of\ intersected\ wetland\ pixels\ in\ validation\ and\ CW\ maps}{Area\ of\ wetland\ pixels\ in\ CW\ map}.$$



SC is calculated at 15 arc-sec resolution by intersecting CWs and validation datasets, and it ranges from 0 to 1 with higher values showing greater similarity between two datasets. For pair-wise comparisons of datasets with different wet fraction, the Jaccard index (JI) is better suited. This index is the fraction of shared wetlands in CW and the validation dataset over the size of their union:

$$JI = \frac{Area\ of\ intersected\ wetland\ pixels\ in\ validation\ and\ CW\ maps}{Area\ of\ wetland\ pixels\ in\ union\ of\ CW\ and\ validation\ maps}.$$

JI ranges from 0 to 1 as well, and a zero index represents the case in which the two datasets are disjoint, and a value of one occurs if two datasets are exactly the same. The last criterion is the spatial Pearson correlation coefficient, further referred to as SPC. SPC is independent from the wet fractions in the CWs and evaluation datasets but is sensitive to the spatial distribution pattern in pair-wise comparisons. SPC values range from 0 to 1 with higher values showing greater similarity. Although the first two criteria were applied for comparison at the original 15 arc-sec resolution, SPC was calculated based on aggregated wetland densities at 0.5° resolution.

Spatial similarity evaluations are displayed as radar charts in Fig. 4 for RFW and the different CW maps for the globe and the selected regions. Because the values of the criteria are sometimes quite similar, three CW maps were selected for display in colour for clarity while the others are shown in grey.

### 4.1.1 Global analysis

With the exception of CW-WTD, which is always more similar to GDW-WTD because the latter is a component of the former, the validation criteria for the CW maps are rather small overall (between 0.2 and 0.6). However, the criteria are larger than the same values between the surface water and wetlands datasets (less than 0.3 in Table 5 for the SPC of the globe and Table S1) showing their advantages. CW maps (especially CW-TCI maps) are more similar to GDW-WTD and Hu's map with respect to GLWD-3 because all but GLWD-3 share the GW modelling methodology. In contrast, the RFW map extends over a 60% larger surface area than GLWD-3 and displays the highest similarity to GLWD-3, suggesting that wetlands in GLWD-3 are the regularly flooded ones. The inclusion of GDWs in the CW maps makes them depart from GLWD-3, but it markedly increases their similarity to the other two validation datasets for JI and SPC (*e.g.,* SPC [RFW, GDW-WTD]=0.3 versus SPC [CW-TCI(15%), GDW-WTD]=0.6). As demonstrated in Fig. 4a, increasing the GDW contribution from CW-TCI(6.6%) to CW-TCI(15%), as an example, also improves the similarity criteria (except the SC for GLWD-3 and GDW-WTD), justifying the need to account for the GDWs to supply a comprehensive description of wetlands.

The following section breaks down the comparative wetland representation between our maps and those of the validation datasets at the regional scale. The selected regions encompass different climates, vegetation covers and ecosystems, both within and outside important wetland areas of the world, to assure the applicability of CW maps. These six regions are France in the temperate climate, the Amazon basin and Southeast Asia over the tropical zone, the cold boreal areas of the Hudson Bay lowlands and Ob river basin, and the Sudd swamp in South Sudan with a semi-arid savannah climate.

### 4.1.2 France

Over France, wetland fractions from the validation datasets are highly inconsistent (Fig. 5). Visible range satellite imagery (JRC surface water) shows the smallest wet fraction (1%). The GLWD-3 and ESA-CCI maps also produce low wetland coverage whereas GIEMS-D15, which essentially forms the RFW map, gives 12% coverage concentrated along the coastline and over the floodplains of the northern rivers. Wetlands from GW modelling-based datasets cover even larger areas (14% and 18% in GDW-WTD and Hu et al., 2017) and are scattered countrywide, except for the French Pyrenees and the Alps, with moderately denser wetlands along large rivers (such as the Rhine floodplain at the eastern border) and the Landes (South-western shore). The MPHFM map (Berthier et al., 2014) can be considered as the most comprehensive validation dataset for the country because it relies on hydromorphic soil properties and was extensively validated. This map shows much larger



wetland extents (23% of France) than the above estimates because of its inclusion of both floodplains (along the Loire, Saône and Rhône rivers) and groundwater-driven wetlands, including those over the weakly permeable granites of Brittany (shown in green in Fig. 5g). These notorious wetlands are not considered in the global datasets but are captured to a good extent in CW maps (Fig. 5).

By combining RFWs (which overlap with 20% of MPHFM) and GDWs, our CW maps capture many features of the MPHFM map, including the total wetland extent (23% for MPHFM versus 22% and 25% for our two CW maps) and correctly capturing most of the coastal and riparian wetlands (Fig. 5). The larger wetland fraction in MPHFM and CW maps is consistent with the work of Pison et al. (2018), who found that (wetland-driven) methane emissions over France deduced from atmospheric inversion were almost a third higher than those simulated by state-of-the-art climate models driven by the global
datasets covered above. The added value of CW maps is demonstrated by the higher similarity criteria between CW maps such as CW-TCI(15%) and MPHFM (SPC=0.52) than between surface water maps such as GIEMS-D15 and MPHFM (SPC= 0.43). However, it is difficult to identify the best CW map over France based on the similarity criteria against MPHFM because four of our CW maps display nearly the same values (Fig. 4b and Table S2).

### 4.1.3 Amazon basin

The Amazon River basin is considered one of the richest tropical wetland ecosystems in the world (Mitsch and Gosselink, 2000). For ease of comparison, we limited our study to the domain of Hess et al. (2015), which covers 5 million km$^2$ (Fig. 6). RFWs (mostly consisting of GIEMS-D15) show a pattern rather similar to that of GLWD-3 and Hess et al. (2015) (Fig. 4c and Fig. 6d, g, h), covering only the main drainage network of the Amazon and certain seasonally flooded wetlands and floodplains. However, certain spatial disagreements exist among these three datasets in seasonally flooded wetlands such as Llanos de
Moxos (12°30'-17°30' S, 63°-68° W), the Roraima savannah, and the Negro River basin (2° N-2° S, 60°-65° W), which are larger in RFW and Hess et al. (2015) than in GLWD-3.

The CW maps capture the wetland pattern of GDW-WTD and Hu et al. (2017) considerably better than RFW (Fig. 6), highlighting the significance of groundwater wetlands over the Amazon. Wetland densities in CW maps, Hu et al. (2017) and GDW-WTD are more realistically high over the leached and swampy soils of the Northern Amazon basin (*e.g.,* Japurá-
Solimões-Negro moist forests) and over the Purus-Madeira ecoregion, in line with recent estimates of wetlands and peatlands. This result suggests that the extended shallow peatlands of South America are the main causal contributor to the global tropical wetland extent (Gumbricht et al., 2017). The higher wetland densities of CW maps with respect to all satellite observations over these particular areas can be attributed to the coincidence of GDWs and dense rainforests (covering almost two thirds of this domain), with large non-flooded wetlands over the Amazon. River channels and surrounding floodplains are better
represented in CW maps, as compared with Hess et al. (2015), due to the inclusion of the RFW component. Similarly, CW compares well against other datasets because almost none of the river floodplains are delineated in Hu et al., 2017, and GDW-WTD misses the Tapajós River floodplain and portions of downstream Amazon corridor. However, CW maps represent the wetland extent in lower density over grassland/savannahs and the Andes dry regions compared with the validation datasets.

### 4.1.4 Southeast Asian deltas

The selected window over South and Southeast Asia stretches over notably wet regions, similar to the Amazon, but with severe human interference and deforestation (Miettinen et al., 2011; Stibig et al., 2013). In Southeast Asia, RFWs (mostly composed of GIEMS-D15) are larger than all validation datasets (Fig. 7d-g) because GIEMS-D15 also detects inundated areas associated with cultivation activities such as rice paddies (Fluet-Chouinard et al., 2015), which are not considered in inventories and GW modelling-based estimates. Over the window, RFWs and CWs coincide with the majority of wetlands in the validation
sets, particularly over the Ganges-Brahmaputra floodplain, Northern Indochina and Yunnan plateau subtropical forests (Fig.



4d: SC between 0.59 and 0.91), showing the good agreement of our developed maps with respect to spatial patterns. As a general rule in Southeast Asia, floodplains and deltas (Ganges, Brahmaputra, Irrawaddy, Mekong and Red Rivers) extend over larger areas in CW maps than in validation maps (Fig. 7), giving a more realistic extent than those in Fan et al. (2013) and Hu et al. (2017) considering the vast flood irrigated cultivation lands along floodplains. However, only a few small wetlands in

the validation datasets are missed in RFW (and CW maps), such as the upstream Mekong River corridor (near 20°N-102°E) and Irrawaddy River (near 25°N-97°E) in GLWD-3.

The CW-WTD and CW-TCI(15%) maps present patterns that are highly similar to each other (Fig. 7h, i) and to the validation datasets. However, high similarity criteria (especially SC) can be the result of large extension of RFWs, itself overlapping almost all of the wetlands in the validation datasets. In addition, the similarity of CW-WTD and CW-TCI(15%),

also derived from similarities between their GDW components, notes that groundwater wetland formation is almost completely explained by topography and climate (of the TCI formulation) in these areas and the negligible role of subsurface characteristics included in GDW-WTD.

### 4.1.5 Hudson Bay lowlands

The Hudson Bay lowlands (HBL) is a vast flat wetland area in the low subarctic regions of North America dominated by

extensive peatlands, swamps and marshes (Mitsch and Gosselink, 2000; Packalen et al., 2014), where below-freezing temperatures for most of the year reduce drainage in the soil layer (Hamilton et al., 1994). A systematic contrast is noted between inundation maps (Fig. 8a-c; maximum wet fraction: 21%) and validation datasets (Fig. 8d-f; minimum wet fraction: 49%) underlining the inability of satellite imagery to capture wetlands in this area (*e.g.,* Landsat images used in JRC surface water, Fig. 8c). Surprisingly, GLWD-3 has a pattern notably similar to those of the other two validation maps due to the

comprehensive wetland maps in Canada available to its developers. CW maps perform fairly well, particularly CW-WTD, which predominantly obtains the highest validation criteria (Fig. 4e). Due to an explicit parameterization of the permafrost (adjusted to reproduce the "observed wetland areas" in Northern America, Fan and Miguez-Macho, 2011), dense wetlands are extended south of 50°N in the GW model by Fan et al (2013), which are less dense in CW-TCI(15%) in the absence of a soil-freezing mechanism. Comparing wetlands detected through satellite imagery and validation datasets, GW modelling appears

to be the best wetland delineation method over boreal zones due to non-permanent surface inundation, shallow WTD or snow/ice cover.

### 4.1.6 Ob River basin

The Ob River basin in western Siberia extends over ~$3 \times 10^6$ km$^2$. The annual variability of the inundated area is large (*e.g.,* Mialon et al., 2005), making this basin one of the largest wetland complexes in the world, which contributes to buffer peak

discharge during the flooding period (Grippa et al., 2005). Wetland fractions in different datasets compare similarly to HBL, except for GLWD-3, which appears to underestimate the total wetland extent although the climatic and geomorphologic properties are nearly alike. Datasets recognizing the contribution of GW to wetland formation (Fig. 9e, f, h, i) indicate consistently higher wet fractions than others. However, CW-TCI(15%) appears to miss wetlands south of 60° N that are extended to the upstream Ob river basin near 50° N in both GDW-WTD and Hu et al. (2017), most likely due to the permafrost

effect on wetland formation. With respect to the evaluation criteria, although CW-TCI(15%) outperforms for SPC, CW-WTD often displays better criteria values. TCTrI-based CW maps fail to surpass others in the validation process, considering that we used the transmissivity map without the permafrost adjustment due to its imprecise representation of hydraulic conductivity in these zones (Sect. 2.4.4). CW-WTD tends to better capture the wetland extent and spatial pattern with more concentrated wetlands in the downstream lowlands and north-western regions (65° N-65° E) of the basin due to RFWs. Overall, considering

the wetland fraction solely attributed to GDWs in CW-TCI(15%) and CW-WTD (13% and 29% of the basin area) and the



difference found between the inundation and validation dataset (Fig. 9, first and second row), it becomes clear that the uncertainty of the wetland extent and spatial pattern is rather high over boreal zones.

### 4.1.7 Sudd swamp

This large wetland is located in eastern South Sudan, nearly 300 m above mean sea level and is the largest freshwater wetland in the Nile basin (Sutcliffe et al., 2016). The Sudd swamp extent estimations are highly uncertain in the literature, ranging from 7.2 to $48 \times 10^3$ km$^2$ (Mohamed et al., 2004 and references therein). Over the selected window, the wetlands and surface water distribution is also highly disparate and varies from 1% to 27% for different datasets (Fig. 10). Additionally, wetlands in Hu et al. (2017) are rather patchy and show sharp density changes with what seems to be periods of 0.5°. Because GLWD-3 appears to represent only flooded wetlands (with the same wetland fraction of RFWs and overlapping with one third of them), and Hu et al. (2017) contains technical issues, GDW-WTD can be considered as the only comprehensive validation dataset over the Sudd swamp.

The CW datasets in Fig. 10 show high wetland densities in the central floodplain, in rather good agreement with GLWD-3, GDW-WTD and regional estimates of saturated soil (compared with visuals in Mohamed et al., 2004; Mohamed and Savenije, 2014). CW-WTD compares more similarly to validation datasets, closely followed by CW-TCI(15%) (Fig. 4g), but
the main difference between these two CW maps is that the groundwater wetlands in CW-TCI(15%) are extended southwest into the southern National Park (over local flat valley bottoms) but are more concentrated over the main floodplain in the SE-NW direction for CW-WTD. The total wetland fraction is nearly equal in CW-TCI(15%) and CW-WTD (25 and 27% of the selected window area), underlining a primary role of topography and climate in wetland formation. Considering the wetland fraction in the RFW map (mostly consisting of ESA-CCI wetlands) and GDW-WTD, groundwater wetlands appear to be the
dominant feature in the Sudd swamp, as is the case for CW-WTD and CW-TCI(15%). The added value of CW maps with respect to GDW-WTD is not substantial, but they additionally contain the seasonally flooded plains downstream of the White Nile (top right of the selected window in Fig. 10g: 12°-14° N, 32-34° E), which are not completely captured by validation datasets due to the inclusion of RFWs.

### 4.2 Wetland extents

Fig. 11 shows that the global wetland fractions of the different CW maps are in range of those in Fan et al. (2013) and Hu et al. (2017), with twice the wetlands in GLWD-3, itself 60% smaller than the RFW extent. Over France (Fig. 11b), the wetland fraction of the CW maps is notably similar to that of MPHFM, which is a calibrated and validated wetland dataset including the GDWs. The regional uncertainty of CW maps is smaller over subtropical areas and higher over boreal and tropical zones. For instance, although the global wetland extents of CW-WTD and CW-TCI(15%) are nearly equal, the former contains 52%
more wetlands over the Hudson Bay lowlands. However, in Southeast Asia, where RFWs have a rather large contribution to total wetlands, CW maps are in relative agreement on wetland extents, whereas the validation dataset appears to critically underestimate the wetland extents. The underestimation of global validation datasets, especially GLWD-3, is quite clear in France, the Amazon and the Ob river basin. Nevertheless, regional differences in wetland fractions among CW maps reaching up to 25% in the HBL and Amazon basin (due to the effect of permafrost in Northern latitudes and high effective precipitation
over the tropics) make our estimates uncertain as well. Additionally, the uncertainty of the reference validation datasets is almost always higher than that of CW maps (global: CW 7%, validation 17%; Ob basin: CW 25%, validation 32%). This uncertainty is driven by several factors, from detection and methodological errors in satellite imagery (flooded extents are reported from 1.4% to 13.2% of the land area in Table 1) to modelling and thresholding uncertainties. Overall, the wetland extents of CW maps are not provable as correct, which is also the case for any wetland mapping efforts at the global scale that
extend the definition of wetlands beyond inundated zones.



## 5 Discussion

### 5.1 Selection of two representative CW maps

If none of the resulting CW maps systematically over-perform the others, two of them usually display the best similarity scores, namely, CW-TCI(15%) and CW-WTD (Fig. 4, Table 5 and Tables S1 to S7 in the supplementary document). These
two datasets (hereafter simply referred to as "CW maps") have many similarities, and by construction, they have almost the same wetland extent (*ca* 21%), and the combination with RFWs reduces the differences found between the corresponding GDWs in boreal and tropical areas (Fig. 3). Both CW maps are among the highest estimates of global wetland, considerably larger than GLWD-3 and close to Hu et al. (2017). An interesting point is that the SPC between these two CW maps and the existing wetland datasets is higher than the SPC among these existing datasets (Table 5), which are rather low (*e.g.,* the SPC
between JRC surface water and GIEMS-D15 is 0.4). This observation underscores that the two outperforming CW maps reconcile the differences between existing wetland maps, whether they focus on RFWs (ESA-CCI, GIEMS-D15 and JRC surface water) or also encompass non-inundated wetlands (GLWD-3, GDW-WTD and Hu et al., 2017).

### 5.2 Zonal patterns

Despite many similarities, the zonal distributions of the CW maps, RFW and validation datasets are sometimes different.
Generally, wetland datasets such as GLWD-3 and GDW-WTD appear to underestimate global wetland extents with respect to CW maps (Fig. 12 and the visuals for France and Southeast Asia: Fig. 5 and 7). The latitudinal patterns are also different among maps in Fig. 12, particularly over the tropics and the boreal zones. Although the wetlands in all validation datasets and CW-WTD are densely concentrated between 50°N and 60°N, in the RFW map, the Northern subtropical (25°N-35°N) and boreal (60°N-70°N) wetlands are of similar extent (1.9 and 2.0 million km$^2$), and in CW-TCI(15%), tropical wetlands (10°N-
10°S) globally outweigh others (covering almost 9 million km$^2$). In fact, tropical wetlands in both CW maps are much more extensive than the maximum reported wetland extents for these latitudes in the literature (almost 5.6 million km$^2$ in Hu et al., 2017). This result is in accordance with recent studies signalling underestimation of tropical wetlands and the subsequent underestimation of their effect on the energy, water and carbon cycles (Collins et al., 2011; Gumbricht et al., 2017; Melton et al., 2013).
Focusing on the differences between CW maps, because the two selected maps are constrained to share the same GDW extent, a trade-off exists between northern and tropical wetlands. In CW-WTD, northern wetlands extend further south into the Sakhalin Taiga and Prairie Pothole Region, as shown by the green belt between 40° and 60°N in Fig. 13c. This southward extension is actually stronger than the permafrost zones (Gruber, 2012), suggesting that the description of the permafrost region in CW-WTD leads to wetland densities that are too strong. However, in the absence of an explicit mechanism for freezing and
permafrost in the TCI formulation, CW-TCI(15%) is prone to underestimating boreal wetlands. Additionally, the difference between the CW maps over the humid tropical zones is consistent with the fact that TCI assumes that effective precipitation is entirely available for wetland formation while it also contributes to surface runoff in the model used by Fan et al. (2013).

### 5.3 Relative role of RFWs and GDWs

Based on the intersection areas between RFWs and GDWs (Table 3) and the global CW fractions, 55% of the global
composite wetlands are solely groundwater-driven, with varying contribution levels in different ecoregions and climate zones. GDWs are the main wetland classes in the tropics and to a lesser extent in the boreal zones. RFWs dominate over the North American lowlands (Fig. 8), Southeast Asia (Fig. 7) and coastal areas and the tropical/subtropical transitional latitudes (Fig. 3c and Fig. 12).





The role of RFWs and GDWs is further analysed in six wetland "hotspots" common to both CW maps (indicated by rectangles in Fig. 13a,b). These areas cover 22% of the land surface area, yet account for 75% of the wetland surface area: (1) North American cold lowlands and permafrost regions, (2) South American tropics and equatorial basins, (3) Ob river basin and west Siberian plains, (4) African northern savanna belt, (5) Wetlands and rice paddies in north-eastern Indian plains and Southeast Asian river deltas, and (6) Coastal wetlands, within a 100 km distance to oceans and with an elevation <100 m above sea level. The total wet fractions in the hotspot windows reach 40% and always exceed the mean global wetland extent (Fig. 14). To ensure that the relative contributions of RFWs/GDWs are meaningful, we tested their sensitivity to the size of the windows. This adjustment had little impact in most areas except for the coastal wetlands, where the wet fraction in both CW maps increases from 43% to 64% when the coastal band is narrowed from 100 km to 20 km. Almost 40% of the RFWs in these areas is located within a 100 km distance to oceans and seas and can be assumed to predominantly represent coastal water bodies (tidal fresh/saline water marshes and river deltas). However, it must be acknowledged that a more rigorous differentiation between coastal wetlands and inland open-water wetlands requires in situ observations or complementary soil and vegetation information.

Outside of the hotspots described above, our CW maps contain small GDWs, ephemeral streams and oases. Such scattered wetlands cover less than 5% of the land area (*ca* 7 million km$^2$ in both CW maps), but they are of great importance for life in semi-arid and arid areas. Many oases and small depressions of this type are represented in CW maps in North Africa, the Arabian Peninsula, southern US and Central Asia and are not captured in any previous mapping efforts, to the best of our knowledge. These bodies are strongly driven by GW and are more difficult to detect by satellite imagery because their size and saturation level change rapidly, sometimes faster than the revisit period of the satellites. As such, we might represent water bodies that cannot be captured by existing satellite-based surveying techniques, but we have not validated these small wetlands against local observational data in this study.

## 6 Data availability and application

The dataset consisting of the two selected composite wetlands maps (CW-WTD and CW-TCI(15%)) is supplied in raster format at 15 arc-sec resolution through PANGAEA https://doi.pangaea.de/10.1594/PANGAEA.892657.Pixels located in oceans and glaciated lands of Greenland are assigned NoData, whereas the remainder of land is split into four classes with distinct codes for non-wetlands, the intersection of GDW and RFW, and "pure" RFW and GDW. All of the datasets used as input to the generation of these dataset were available via open access for research and educational applications and could be accessed through the web links mentioned in their accompanying scientific papers.

These classified maps are believed to be useful for hydrological or land surface modelling by assigning specific properties or processes to the places identified as wetlands or floodplains. As such, the CWs can be viewed as the spatial support of a particular "hydrotope" (Gurtz et al., 1999; Hattermann et al., 2004), *i.e.,* the hydrological analogue of plant functional types (PFTs) for vegetation properties and processes (Lafont et al., 2012). Potential applications include estimation of methane production or denitrification by wetlands, especially if combined with dynamic modelling of the saturation degree within the wetland fractions (Hesse et al., 2008; Post et al., 2008).

## 7 Conclusions and perspectives

Although well-known regional-scale wetlands have been thoroughly inspected and their geographic and temporal properties have been analysed in many areas of the globe, no universal agreement exists among datasets or estimates for the total global extent and spatial distribution of wetlands. Despite near-global coverage of wetland maps based on satellite imagery, most of them only locate inundated areas and overlook non-inundated groundwater-driven wetlands. To develop a



comprehensive global wetland dataset, we merged regularly flooded wetlands (RFWs) and groundwater-driven wetlands (GDWs) to develop CW maps. The corresponding maps were produced globally at high resolution and subsequently overlapped to form the CW maps under the assumption that they are all relevant although not exhaustive. The same applies to the three maps of flooded areas derived from satellite imagery that were overlapped to construct the RFW map. In contrast,

we did not end up with a single GDW map because of several uncertainties related to this component.

Whether derived from simplified or direct WTD modelling (based on the wetness indices or on the estimates from Fan et al., 2013), a major challenge is to define thresholds on TI or WTD to separate the wet and non-wet pixels. Following the existing literature, wetlands were defined as areas where the mean WTD is less than 20 cm, and this WTD threshold was translated into the TI threshold defining the same global wetland extent (15%). These choices necessarily remain subjective in

the absence of consensual global wetland map and definition, and the related uncertainty in wetland extent was shown to amount to a few percent of the total land area based on sensitivity analyses for reasonable values of the different thresholds. We also considered several classic variants of the TI to conclude that the TCI (topography-climate wetness index), also favoured by Hu et al. (2017) with a modified formula, offers the best correspondence with the validation datasets. The original TI did not capture the wetland density contrasts between arid and wet areas, and the inclusion of sub-surface transmissivity in

TCTrI induced overly sharp density contrasts that did not always match the recognized patterns of large wetlands. This situation calls for improved global transmissivity datasets or new methods to supply a more continuous description of transmissivity than those currently proposed based on discrete classes of lithology (Hartmann and Moosdorf, 2012; Gleeson et al., 2014) or soil texture (Fan et al., 2013).

The validity of the composite maps is mostly supported by the good match with the MPHFM dataset developed by Berthier

et al. (2014) over France because it was tailored to comprehensively include flooded and non-flooded wetlands with calibration against hydromorphic soils and validation against local surveys. Comparisons with other validation datasets at the global and regional scales further show that the composite maps capture the main wetland patterns over the globe. This process also led us to select two composite maps, namely, CW-WTD and CW-TCI(15%), showing the best overall match with the validation datasets. Wetlands in these maps respectively cover 27.5 and 29 million km$^2$, *i.e*., 21.1 and 21.6% of the global land area

(excluding lakes, Antarctica and the Greenland ice sheet). These wetland fractions are close to the maximum of the literature range, together with recent estimates also recognizing the contribution of groundwater-driven wetlands (Fan et al., 2013; Hu et al., 2017).

All these estimates, including ours, overlook the loss of wetlands induced by anthropogenic pressures, which is estimated to approach 30% of undisturbed or potential wetlands (Sterling and Ducharne, 2008; Hu et al., 2017), mostly due to

urbanization and agricultural drainage. This situation is especially true for GDWs because most human influences on the environment were neglected in the input datasets (climate, topography, transmissivity, and sea level) for global WTD modelling. In contrast, the RFW map was derived by overlapping satellite imagery for the contemporary period (past 5 to 34 years), thus showing most human-induced changes on the surface water (including the way in which damming shifts wetlands to lakes or drylands, as reported by Pekel et al., 2016). Nevertheless, the overlap of several inundation datasets with different

historical depths was intended to minimize these disturbances, as justified by the higher spatial correlation between the inundation datasets and the CW maps than between themselves.

In this framework, an important conclusion is the marked similarity between the two proposed composite maps, despite their different assumptions for GW modelling. In particular, both maps locate 75% of the global wetlands within six wetland hotspot regions, in boreal and tropical areas and along the shoreline (coastal wetlands). Higher wetland densities in the tropics

compared with other datasets originate from the GDW contribution in regions with dense canopy and/or cloud cover. These conditions are tightly linked in the humid tropics, where wetlands have long been underrepresented (Collins et al., 2011; Melton et al., 2013; Gumbricht et al., 2017). The largest differences between the two proposed CW datasets are found in the boreal zones (including the two hotspots of the Prairie Pothole Region and East Siberian Taiga), although the RFWs are the





dominant components. This uncertainty corresponds to subsurface conditions (transmissivity) and might be reduced having a better and higher-resolution description of the permafrost extent, active layer depth, hydraulic conductivity, or organic matter content.

Another major feature of the two composite maps is the importance of small and scattered wetlands, as shown by the
extent of wetlands outside the six hotspots (3.8% to 5.2% of the land area according to CW-WTD and CW-TCI(15%), respectively). This is yet another feature derived from the GDWs because these small wetlands are often difficult to detect using satellite imagery techniques, especially for the non-inundated or ephemeral wetlands with sizes that vary rapidly compared with the revisit period of the satellites. The resolution used in this work (~500 m at the Equator) is sufficiently fine to detect many of these small wetlands but still coarser than patchy wetlands in small depressions or riparian zones along
small-order streams. This approach might lead to overestimation of the extent of these small wetlands because the pixels are either fully wet or fully non-wet. A better delineation is expected from higher resolution DEMs, such as the MERIT (Multi-Error-Removed Improved-Terrain) DEM of Yamazaki et al. (2017), offering a worldwide 3-arc resolution.

By distinguishing the RFWs and GDWs, the proposed datasets eventually offer a simple wetland classification focused on their hydrologic functioning. Compared to classic wetland classifications, which are strongly based on floristic inventories
and habitat typologies (*e.g.,* Zoltai and Vitt, 1995; Finlayson et al., 1999; Lehner and Döll, 2004; Herold et al., 2015), we separated areas where wet conditions at the surface are primarily driven by flooding or GW inputs or both, where the two classes intersect. Since the underlying principles and input datasets are globally valid, this classification is believed to be highly useful for land surface hydrological modelling. In particular, we intend to use this process in the ORCHIDEE land surface model (Krinner et al., 2005; Ducharne et al., 2017) to describe the areas where GW convergence from the uplands to the
lowlands can lead to high soil moisture, with a potential to enhance the local evapotranspiration and related land-atmosphere feedback (*e.g.,* Bierkens and van den Hurk, 2007; Maxwell et al., 2007; Vergnes et al., 2014; Wang et al., 2018).

### Acknowledgment

This research is a part of the PhD project of Ardalan Tootchi, funded by "Région Ile de France" via the "Réseau francilien de recherche sur le développement soutenable".

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





**Table 1: Summary of water body, wetland and related proxy maps and datasets from the literature. The wet fractions indicated in % in the last column are those indicated in the reference paper or data description for each study.**

| Name and reference | Resolution | Type of acquisition | Wetland extent | |
|---|---|---|---|---|
| | | | (million km²) | % of the land* |
| Maltby and Turner (1983) | - | Based on Russian geographical studies | 8.6 | 6.6% |
| Matthews and Fung (1987) | 1 degree | Development from soil, vegetation and inundation maps | 5.3[†] | 4.0% |
| Mitsch and Gosselink (2000) | Polygons | Gross estimates, Combination of estimates and maps | ~20[†] | ~15.3% |
| GLWD-3 Lehner and Döll (2004) | 30 arc-sec ~1km | Compilation of national/international maps | 8.3 - 10.2[‡] | 6.2 - 7.6% |
| GLC2000 Bartholomé and Belward (2005) | 1 km at Equator | SPOT vegetation mission satellite observations | 4.9 | 3.4% |
| GIEMS Prigent et al. (2007) | 0.25° ~25km | Multi sensor: AVHRR, SSM/I, Scatterometer ERS | 2.1 – 5.9 | 1.4 – 4% |
| Fan et al. (2013) | 30 arc-sec ~1km | Groundwater modelling | ~19.3[†] | ~17% |
| GLOWABO Verpoorter et al. (2014) | Shapefiles of lakes larger than 0.002 km2 | Satellite imagery: Landsat and SRTM topography | 5 | 3.7% |
| SWAMPS Schroeder et al. (2015) | 25 km | Modeling using multi sensor info: SSM/I, SSM/S, QuikSCAT, ASCAT | 7.7 – 12.5[§] | 5.2 – 8.5% |
| ESA-CCI land cover Herold et al. (2015) | 10 arc-sec ~300m | Multi sensor: SPOT vegetation, MERIS products | 6.1 | 4.7% |
| GIEMS-D15 Fluet-Chouinard et al. (2015) | 15 arc-sec ~460m | Multi-sensor: SSM/I, ERS-1, AVHRR, Downscaled from a 0.25° wetland map | 6.5 – 17.3 | 5.0 - 13.2% |
| G3WBM Yamazaki et al. (2015) | 3 arc-sec ~90m | Satellite imagery: Landsat | 3.2 | 2.5% |
| JRC Surface water Pekel et al. (2016) | 1 arc-sec ~30m | Satellite imagery: Landsat, including maximum water extent and interannual occurrence | 2.8 – 4.4 | 2.1 - 3.4% |
| HydroLAKES Messager et al. (2016) | Shapefiles of lakes larger than 0.1 km2 | Multiple inventory compilation including Canadian hydrographic dataset and SWBD | 2.7 | 1.8% |
| Hu et al. (2017) | 1 km | Development based on topographic wetness index and land-cover | 29.8[¶] | 22.5% |
| Poulter et al. (2017) | 0.5° ~50km | Merging SWAMPS and GLWD-3 | 10.5 | 7.1% |

* Percentages are those from the corresponding journal article or book. If no mention of percentage coverage exists, the value is calculated by dividing the wetland area by the land surface area excluding Antarctica, the glaciated Greenland and lakes.
† Excluding Caspian sea and large lakes
‡ Excluding Antarctica, glaciated Greenland, including lakes and Caspian sea. Additionally the range in GLWD is different based on interpretation of fractional wetlands.
§ Excluding large water bodies
¶ Including the Caspian sea



**Table 2: Layers of wetlands constructed in the paper, their definitions and the subsections where they are explained. Total land area for wetland percentages excludes lakes, Antarctica and the Greenland ice sheet.**

| Layer | | | Definition | Wetland percentage | Explained in |
|---|---|---|---|---|---|
| **RFW** (Regularly Flooded Wetlands) | | | Union of three inundation datasets (ESA-CCI, GIEMS-D15, JRC surface water) | 9.7% | Sect. 3.1 |
| **GDW** (Groundwater Driven Wetland) | WTD | | Pixels with water table depth less than 20 cm (Fan et al. 2013) | 15% | Sect. 3.2.1 |
| | TI | (6%) | Pixels with highest TIs, covering 15% of total land when combined with RFW | 6% | Sect. 3.2.2 |
| | | (15%) | Pixels with highest TIs values covering 15% of land | 15% | |
| | TCI | (6.6%) | Pixels with highest TCIs, covering 15% of total land when combined with RFW | 6.6% | |
| | | (15%) | Pixels with highest TCI values covering 15% of land | 15% | |
| | TCTrI | (6%) | Pixels with highest TCTrI, covering 15% of total land when combined with RFW | 6% | |
| | | (15%) | Pixels with highest TCTrI values covering 15% of land | 15% | |
| **CW** (Composite Wetland) | WTD | | Union of RFW and GDW-WTD | 21.1% | Sect. 3.3 |
| | TI | (6%) | Union of RFW and GDW-TI(6%) | 15% | |
| | | (15%) | Union of RFW and GDW-TI(15%) | 22.2% | |
| | TCI | (6.6%) | Union of RFW and GDW-TCI(6.6%) | 15% | |
| | | (15%) | Union of RFW and GDW-TCI(15%) | 21.6% | |
| | TCTrI | (6%) | Union of RFW and GDW-TCTrI(6%) | 15% | |
| | | (15%) | Union of RFW and GDW-TCTrI(15%) | 22.3% | |



**Table 3: ArcMap tools used in this study for data processing and their equivalent open-source software.**

| ArcMap | Open-source software | Application |
|---|---|---|
| Polygon to raster (conversion toolbox) | Rasterize (vector to raster) | To convert vector data into raster pixels |
| Project raster (Data management toolbox) | QGIS: Warp (reproject) | Projecting different layers coordinate system to WGS84 |
| Resample & Aggregate (Data management toolbox) | QGIS: Raster calculator | To change the resolution of the rasters |
| Raster calculator (Spatial analyst toolbox) | QGIS: Raster calculator | To intersect/overlap raster datasets |
| Reclassify (Spatial analyst toolbox) | QGIS/GRASS: r.reclass | To merge raster datasets or mask them |

**Table 4: Percent of overlap between GDW and RFW (percent of total land pixels).**

| Groundwater-driven wetland layer | Intersecting with RFW | Non-intersecting with RFW |
|---|---|---|
| GDW-TI(6%) | 0.7% | 5.3% |
| GDW-TCI(6.6%) | 1.3% | 5.3% |
| GDW-TCTrI(6%) | 0.7% | 5.3% |
| GDW-TI(15%) | 2.5% | 12.5% |
| GDW-TCI(15%) | 3.6% | 11.4% |
| GDW-TCTrI(15%) | 2.4% | 12.6% |
| GDW-WTD(15%) | 3.8% | 11.2% |



**Table 5: Correlation between the developed and reference datasets (wetland fractions in 3 arc-min grid-cells). The highest three values in each column are shown in bold format, and grey cells give the values used in Fig. 4.**

| Dataset name | ESA-CCI | GIEMS-D15 | JRC surface water | RFW | GLWD-3 | GDW-WTD | Hu et al. (2017) |
|---|---|---|---|---|---|---|---|
| GDW-TI(15%) | -0.07 | 0.11 | 0.03 | 0.04 | 0.23 | 0.18 | 0.31 |
| GDW-TCTrI(15%) | -0.04 | -0.01 | -0.10 | 0.01 | 0.17 | 0.26 | 0.26 |
| GDW-TCI(15%) | 0.12 | 0.24 | 0.03 | 0.23 | 0.23 | **0.53** | 0.33 |
| GDW-WTD | 0.27 | 0.29 | 0.07 | 0.30 | **0.36** | 1.00 | **0.45** |
| CW-TI(6%) | 0.56 | 0.59 | **0.44** | **0.91** | 0.21 | 0.34 | 0.33 |
| CW-TCTrI(6%) | 0.49 | 0.59 | 0.43 | **0.78** | 0.24 | 0.43 | **0.40** |
| CW-TCI(6.6%) | 0.58 | 0.64 | 0.40 | **0.80** | 0.26 | 0.52 | 0.31 |
| CW-TI(15%) | 0.63 | 0.60 | 0.28 | 0.57 | 0.31 | 0.40 | 0.32 |
| CW-TCTrI(15%) | 0.55 | 0.45 | 0.36 | 0.51 | 0.32 | 0.38 | 0.28 |
| CW-TCI(15%) | **0.70** | **0.71** | **0.47** | 0.69 | 0.28 | **0.58** | 0.35 |
| CW-WTD | **0.63** | **0.69** | 0.37 | 0.65 | **0.34** | **0.65** | **0.43** |
| ESA-CCI | 1.00 | 0.33 | **0.66** | 0.53 | 0.28 | 0.27 | 0.27 |
| GIEMS-D15 | 0.33 | 1.00 | 0.36 | 0.67 | 0.26 | 0.29 | 0.20 |
| JRC surface water | **0.66** | 0.36 | 1.00 | 0.40 | 0.07 | 0.07 | 0.07 |
| RFW | 0.53 | **0.67** | 0.40 | 1.00 | **0.38** | 0.30 | 0.22 |
| GLWD-3 | 0.28 | 0.26 | 0.07 | 0.26 | 1.00 | 0.36 | 0.33 |
| Hu et al. (2017) | 0.27 | 0.20 | 0.07 | 0.22 | 0.33 | 0.45 | 1.00 |



**Figure 1: Density of lakes, regularly flooded wetlands and components of the latter (percent area in 3 arc-min grid-cells). For zonal wetland area distributions (right side charts), the area covered by wetlands in each 1° latitude band is displayed.**







**Fig. 2: Density of scattered groundwater wetlands based on different approaches (percent area in 3 arc-min grid-cells). For zonal wetland area distributions (right side charts), the area covered by wetlands in each 1° latitude band is displayed.**

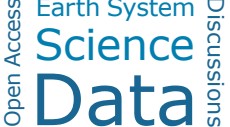

**Fig. 3 Latitudinal distribution of different wetland maps; (a,b) GDWs, (c) components of CW-TCI(15%) and their intersection, (d,e) CWs. The wetland areas along the y-axis are surface areas in each 1° latitudinal band.**





**Figure 4: Evaluation criteria between composite wetland maps and evaluation datasets (a) global scale, (b) France, (c) Amazon, (d) South-East Asia, (e) Hudson Bay Lowlands, (f) Ob basin, (g) Sudd swamp**





**Figure 5: Maps of wetlands in France according to different water and wetland datasets: (a, b, c) components of RFW, (d, e, f, g) validation datasets, (h, i, j) datasets generated in this study. The panels also give the mean areal wetland fraction of each dataset in the study area (using the mean fraction of each fractional wetland class of GLWD-3, cf. Sect. 2.5.1). The bounds of the study are the French metropolitan boundaries.**

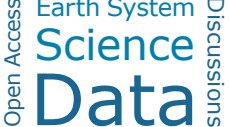



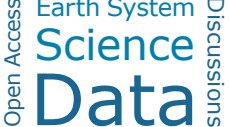

**Figure 6: Maps of the Amazon River basin wetlands according to different water and wetland datasets: (a, b, c) components of RFW, (d, e, f, g) evaluation datasets, (h, i, j) datasets generated in this study. The panels also give the mean areal wetland fraction of each dataset in the study area (using the mean fraction of each fractional wetland class of GLWD-3, cf. Sect. 2.5.1). The bounds of the basin are taken from Hess et al. (2015).**



**Figure 7: Maps of the South-East Asian wetlands according to different water and wetland datasets: (a, b, c) components of RFW, (d, e, f) evaluation datasets, (g, h, i) datasets generated in this study. The panels also give the mean areal wetland fraction of each dataset in the study area (using the mean fraction of each fractional wetland class of GLWD-3, cf. Sect. 2.5.1). The bounds of the study window are (5°-28°N, 82°30'-108°E).**

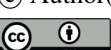





**Figure 8: Maps of the Hudson Bay Lowlands wetlands according to different water and wetland datasets: (a, b, c) components of RFW, (d, e, f) evaluation datasets, (g, h, i) datasets generated in this study. The panels also give the mean areal wetland fraction of each dataset in the study area (using the mean fraction of each fractional wetland class of GLWD-3, cf. Sect. 2.5.1). The bounds of the study area are (48°-56°N, 76°-86°W).**







**Figure 9: Maps of the Ob River basin wetlands according to different water and wetland datasets: (a, b, c) components of RFW, (d, e, f) evaluation datasets, (g, h, i) datasets generated in this study. The panels also give the mean areal wetland fraction of each dataset in the study area (using the mean fraction of each fractional wetland class of GLWD-3, cf. Sect. 2.5.1). The bounds of the basin are taken from the HydroBASINS layer of HydroSHEDS**





Figure 10: Maps of the Sudd swamp wetlands according to different water and wetland datasets: (a, b, c) components of RFW, (d, e, f) evaluation datasets, (g, h, i) datasets generated in this study. The panels also give the mean areal wetland fraction of each dataset in the study area (using the mean fraction of each fractional wetland class of GLWD-3, cf. Sect. 2.5.1). The bounds of the study area are (4°30'-14°N, 24° 30'-34°E).




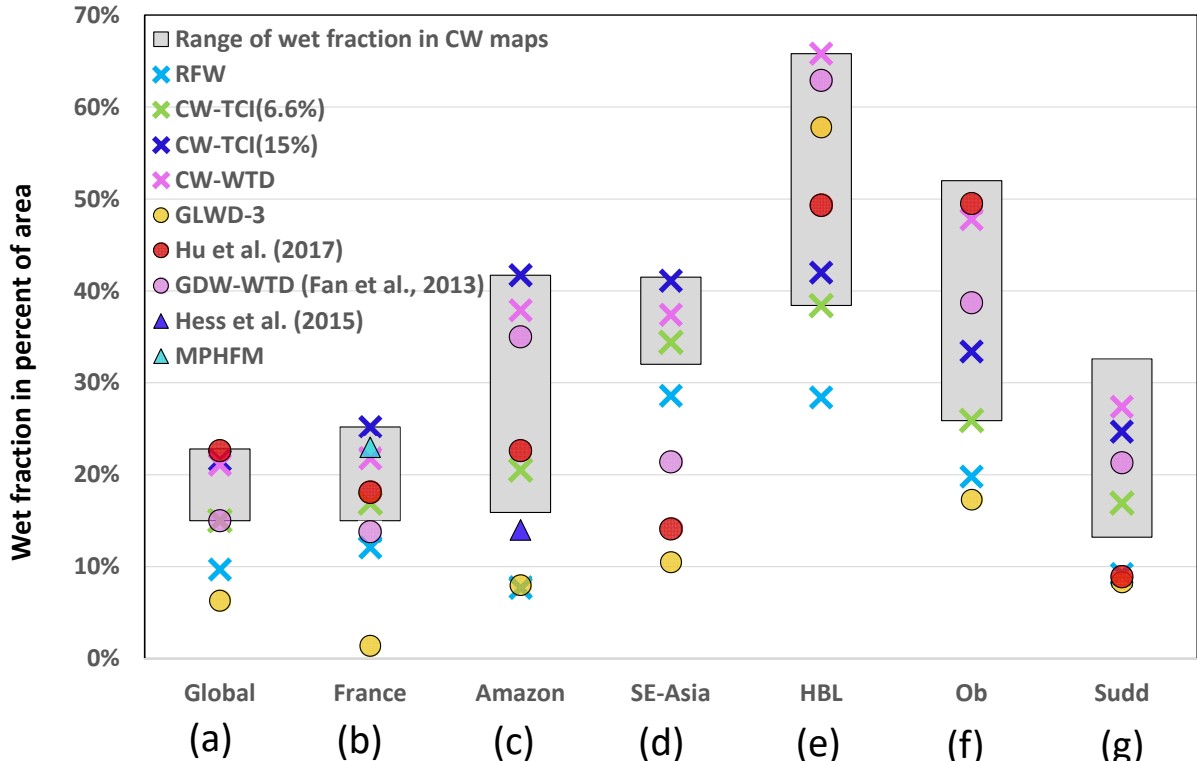

**Figure 11: Total wet fractions for RFW, different CW and validation datasets, at global scale and in the studied regions (values in percent of the corresponding land surface area). Only three CW maps are shown in colours, and other are displayed with the grey range.**



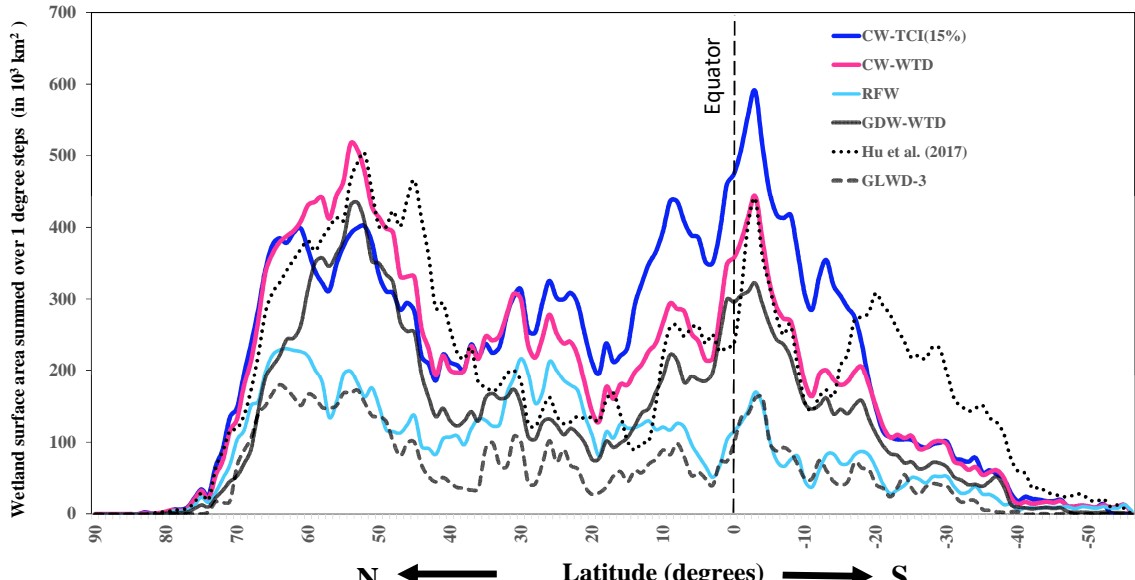

**Figure 12 Latitudinal distribution of the selected CWs and evaluation datasets. The wetland areas along the y-axis are surface areas in each 1° latitudinal band.**



**Figure 13: Wetland density (as percent area in 3 arc-min grid-cells): (a) in CW-WTD, (b) in CW-TCI(15%), (c) difference between them. Numbers on (a) and (b) refer to the wetland hotspot windows explained in Sect. 5. For zonal wetland area distributions (right side charts), the area covered by wetlands in each 1° latitude band is displayed.**





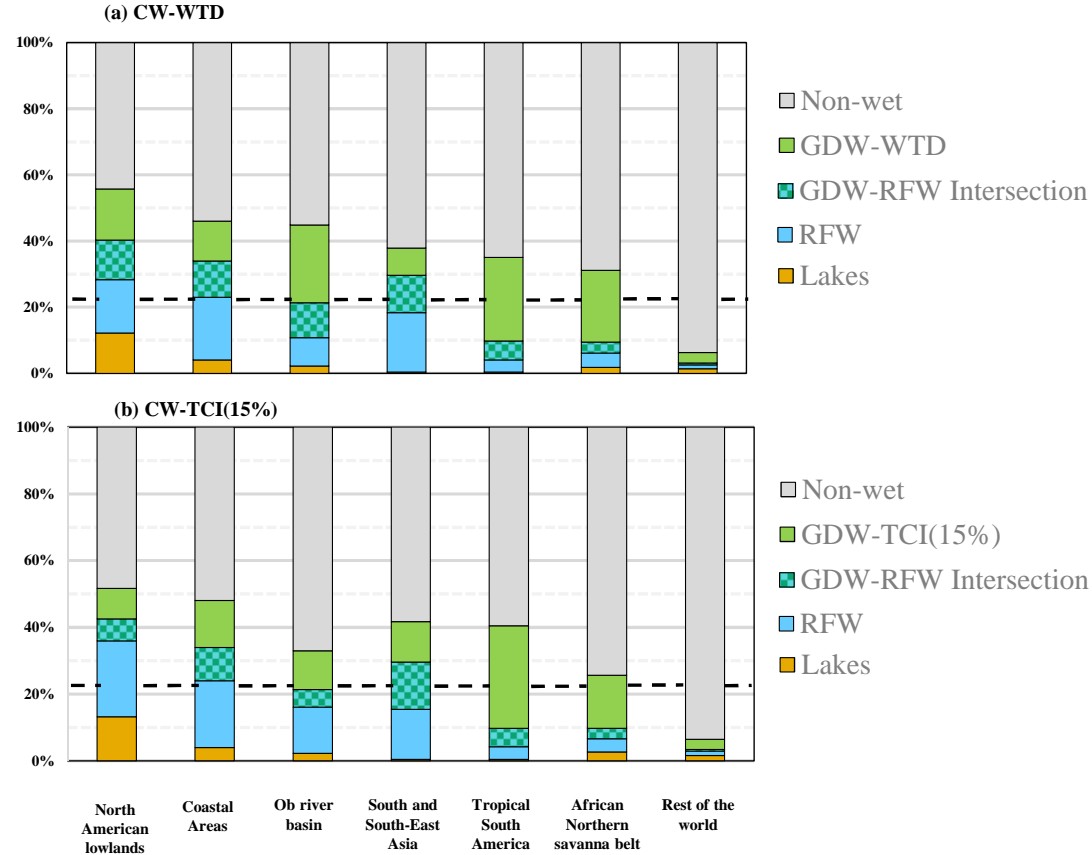

**Figure 14: Contribution of non-wet areas, lakes, RFW, GDW, and their intersection in the wetland hotspot window shown in Fig. 13: (a) in CW-WTD, (b) in CW-TCI(15%). The dashed line shows the average global wetland fraction, equal to 21.1% in (a) and 21.6% in (b).**