# Peer review of "Multi-source global wetland maps combining surface water imagery and groundwater constraints"

_Earth System Science Data, 2018_

## Referee Comment (RC1) · Anonymous Referee #1 · 19 Sep 2018

This paper describes a new global, high-resolution (15 arc-seconds) mapping of wetlands based on hydrological functioning, combining existing surface water imagery and model-derived estimates of groundwater-fed saturated areas. One aim of this new composite dataset (or, better said, the different versions proposed) is to reconcile the large discrepancies existing between current databases of wetlands extent and location. Another is to propose a readily-available datasets for land surface hydrological modeling, and indeed such datasets are critically needed to constrain (eco)hydrological simulations often seeking an increasing level of details. This new dataset is thoroughly evaluated against other global to region-specific datasets, showing various degrees of agreement but with superior overall performance as compared to evaluating existing datasets against each other. In particular, this new estimates shows significant agreement with a specifically-developed dataset fro France where groundwater contribution is also included. In addition, the broad definition used for wetlands here yields one the highest reported estimates for global wetland coverage. In particular, the authors suggests that including groundwater contribution to better captures small scattered wetlands and much-underestimated tropical wetlands.

I found the manuscript pleasant to read, with a clear presentation of the methodology and results, for the most part. I only have a few minor comments, following which I would recommend this work for publication in Earth System Science Data.

**General comments**

- While Sect. 2 is termed "Methods and data", it mostly presents datasets and Sect. 3 presents most of the methodology. I would suggest to move Sect. 2.6 to Sect. 3 and rename Sect. 2 "Datasets" or "Data".

- Several times in text, the authors highlight the dispersion existing between the three datasets of regularly flooded wetlands (RFW) they use. For example, in P9L7 they show that their overlap is only 5% of the summed RFW area coverage. This intuitively raises questions about the accuracy of each of these RFW datasets. Section 3.1.2 somewhat describe some reasons why some features are specifically captured by each one of them, but do not discuss why they do not overlap. It seems to me that such a short discussion would be desirable, given that these datasets are merged to produce the RFW part of the new composite wetland (CW) estimates presented in this study.

- It would be helpful for the reader to draw rectangles corresponding to the sub-regional foci where the authors discuss some of the feature of the datasets, e.g. in Fig. 6 (P13L20-21), Fig. 7 (P14L5-6), and in Fig .10 (L15P22).

- The conclusion is quite long and sometimes feels like a discussion. In addition, it tends to reformulate the main results (e.g. P18L22-27) instead of providing conclusive remarks and outlook. Please consider avoiding too much repetition from Sect 3, and passing elements of the Conclusions over to the Discussion.

**Specific comments**

**P6, L34:** Can the authors explain why they decided not to consider the permafrost effect for $K_s$ estimates? Especially given the likely influence on the performance of TCTrI.

**P10, L16:** The transmissivity ($T_r$) map used here using GLHYMPS is at a much coarser resolution ($1^o$) than the TI map (15 arc-sec). This mismatch is in addition likely to have a larger impact on dataset accuracy of TCTrI than the mismatch present in TCI ($P_e$ is at $0.5^o$), since intra-grid $T_r$ variability, from fine-scale heterogeneities, is likely higher than that of $P_e$. In a way, the lesser performance of TCTrI (or at least its little added value as compared to TCI) was thus somewhat predictable?

**P13, L10:** "covered above" is vague, can the authors specify which datasets they are mentioning?

**P13, L25:** "in line with recent estimates of wetlands and peatlands": references to published works would be needed here.

**Technical comments**

**P6, L34:** The symbol or notation after "polygons of" is not properly rendered.

**P13, L4:** "Fig. 5j", instead of "Fig. 5".

**P15L17** Instead of "CW-TCI(15%)", "CW-TCI(6.6%)", "CW-TI(6%)" , dropping the "%" in the name (CW-TCI15, CW-TCI6.6, etc.) could help avoiding confusion when these names are used to describe wetland extent.

---

## Referee Comment (RC2) · Anonymous Referee #2 · 9 Nov 2018

The paper describes static global data sets (global raster maps) of wetlands showing, with a spatial resolution of 15 arc-sec, the location of regularly flooded wetlands RFW (one map) and of a groundwater-dependent wetlands GDW or rather shallow groundwater table areas (various alternative map versions). According to the authors, the dataset is to be applied in large-scale land surface modelling (hydrological, ecological and biogeochemical modeling) and environmental planning.

General comments

The data set is accessible via the given identifier and is documented. It is significant – unique, useful, and complete. It is unique and complete in that it combines RWF and GWD. It is useful as there is currently a low level of knowledge with respect of the location and size of wetlands at the global scale and there exists a demand for an

improved knowledge. The data set usable in its current format and size.

The article itself is not yet appropriate to support the publication of the data sets.

A) In particular, the uncertainty of the data is not yet discussed enough (read more details below).

B) The authors should explain more in detail how this data set could be used for what purpose, dependent on the (large) uncertainties. Particularly difficult to utilize is the information on GDW. How could, for example, the estimated GWD distributions that are based on simple steady-state modeling of groundwater tables, without interaction with surface water bodies and without taking into account human activities, beactually used in large-scale land surface modelling? Maybe focus on results for France, e.g. Landes.

C) Please state very clearly that the wetlands identified in this data set do not correspond to wetlands with typical wetland vegetation.

D) Validation of the data set is not done correctly. I think it is not appropriate to use GDW-WTD derived from Fan et al. (2013) as validation data set as this data set is the basis of your estimates of GDW in all CW maps, directly or indirectly. It is not correct to say that the other CW maps (those were topo indices were used) are independent from GDW-WTD as the total area of GDW (15% of land area) that is only distributed via the topo indices is prescribed by GDW-WTD. Do not use GDW-WTD as a validation data set.

Regarding a clearer presentation of uncertainty, please discuss 1) the uncertainties of the GIEMS-D15 (and in particular the underlying GIEMS data set) and 2) the GDW data set that, with the resulting 15% of global land area being identified as GDWs by global groundwater modeling (Fan et al. 2013), also constrains the topographic index-derived GDWs.

Regarding 1), the work of Adam et al. (2010) indicates that GIEMS overestimates

inundation area in the Netherlands and Northern Germany and more so in India (if rice paddies are taken into account), likely due to confounding wet soils with inundated areas. This was later supported by unpublished work (Master thesis of Matthias König). Here, the author compared Landsat ETM+ scenes with GIEMS and found a strong overestimation of inundation extent by GIEMS in Ganges-Brahmaputra and Parana river basins during and after months with more than 100 mm rainfall. Evaluating the presented RFW map for the Netherlands and northern Germany (around Hamburg), I do not find it realistic that so many km2 of land are inundated on average in one out of twelve months. I suggest providing the dataset on Google Earth such that the validity/uncertainty of the data set can be judged more easily based on local knowledge about e.g. the mean annual maximum extent of inundation that was predominantly used to produce RFW.

Regarding 2), I would think that with the steady-state groundwater flow modeling approach of Fan et al. (2013), an overestimation of areas with shallow groundwater tables is very likely. In this approach, the location of surface water tables is not taken into account but groundwater flows out freely and is removed if the groundwater table exceeds the land surface elevation. However, rivers are in most cases incised into the land surface, i.e. the river channel forms a long depression as compared to the surrounding floodplain such that the groundwater table will be lowered in comparison to the Fan et al. (2013) approach because the drainage level is lower.

It would be interesting to better understand the map of potentially wet zones in France. With 23% of France being wetland, or rather a potentially wet zone, is this based on having 23% of France covered by soils with hydromorphic features? Do you have an idea about the actually wet area in France, taking into account human impact? Are there maps for France for RFW? Commissioning error in Berthier et al. (2014) is high (75% of validation points are wrongly classified), what does this mean for your comparison?

Specific comments

P4L29ff: Why did you exclude only natural lakes and not also reservoirs that can are estimated in the GRanD database to cover 305.000 km2? Please mention.

P6L6: Modelled WTD was not constrained by observations by Fan et al. (2013) but only compared.

P7L9-10 and Table 1: GLWD-3 values listed in Table 1 actually do not include lakes and reservoirs, and not the Caspian Sea (see Table 3 in Lehner and Döll 2004). Why do the values in the table differ from the values on page 7?

Please explain more clearly in the manuscirpt why the 15% assumption works better than the 6% assumption, by referring more strongly to Table S1.

Technical comments

P6L16: 1-meter DEM?

P16L33: illegible

References Adam, L., Döll, P., Prigent, C., Papa, F. (2010): Global-scale analysis of satellite-derived time series of naturally inundated areas as a basis for floodplain modelling. Adv. Geosci., 27, 45-50. doi:10.5194/adgeo-27-45-2010.

---

## Author Comment (AC1) · 7 Dec 2018

We thank the Referee #1 for his/her comments and suggestions. They will help improve the manuscript. We will address comments in the order discussed by the Referee starting with "General comments". In the following, the Referee's comments are in black, our replies in blue and the options that we considered to change the manuscript in green. New phrases added to text are underlined.

**GENERAL COMMENTS**

**Comment 1:** "While Sect. 2 is termed "Methods and data", it mostly presents datasets and Sect. 3 presents most of the methodology. I would suggest to move Sect. 2.6 to Sect. 3 and rename Sect. 2 "Datasets" or "Data".

**Reply:** We agree with the referee. We will restructure Sect. 2 and Sect. 3 in order to enhance the coherence of the paper. The paragraphs listed below and sections will be moved to Sect.3 and the names of subsections, changed accordingly for more clarity.

**Modifications:**

- 2. Methods and data → *2. Datasets*
- 2.1 Wetland definition and general mapping strategy → *2.1 Mapping strategy and requirements*
  The first two paragraphs of 2.1 will be moved to the new *Sect. 3.1.1 Wetland definition.*
  P4L20-L25: *"In this process, many layers were developed and are summarized in Table 2 and detailed in Sect. 3. The map and methods to exclude lakes from all layers is explained in Sect. 2.2. Input datasets to RFWs and GDWs are presented in Sect. 2.3 and 2.4 respectively, and several independent validation datasets, global and regional, are presented in Sect. 2.5. It should be noted that in the remainder of this paper, the wetland percentages of the land surface area always exclude lakes (Sect. 2.2), the Caspian Sea, the Greenland ice sheet and Antarctica (unless otherwise mentioned). For this reason, these percentages and areas might be different from those shown in Table 1, which are indicated in each original paper or data description.*
- 2.6 Data Processing → *3.1.2 Data processing*

- *3.1 Definitions and layer preparations*
- *3.1.1 Wetland definition*

**Comment 2:** "Several times in text, the authors highlight the dispersion existing between the three datasets of regularly flooded wetlands (RFW) they use. For example, in P9L7 they show that their overlap is only 5% of the summed RFW area coverage. This intuitively raises questions about the accuracy of each of these RFW datasets. Section 3.1.2 somewhat describe some reasons why some features are specifically captured by each one of them, but do not discuss why they do not overlap. It seems to me that such a short discussion would be desirable, given that these datasets are merged to produce the RFW part of the new composite wetland (CW) estimates presented in this study."

**Reply:** It is true that we emphasized more on the disparities between the inundation datasets and we do not discuss them in 3.1.2 (now 3.2.2). We will modify the mentioned section, adding a short discussion on the reasons of disagreement between RFW datasets. But generally, the input datasets to RFW are selected to combine different data acquisition methods. In places where they do not agree (like over Western

Europe and the Ob river basin: Fig R1 and R2) it is because the vegetation cover does not allow to access the soil surface for remote sensing techniques. Globally, RFWs cover 9.7% of the Earth surface while the area where two datasets agree covers only 1.8% of the Earth. The small size of intersection between the three dataset however is attributed mainly to JRC surface water extent which is very small (1.5% of the Earth surface area).

We will also add a subsection (5.1) to discuss the uncertainty of the CW maps underlying layers.

**Modifications:** Underlined parts are the added phrases

- Sect. 3.1.2 (now 3.2.2) first paragraph: "*Overall, the RFW map covers 9.7% of the land surface area (12.9 million km2) including river channels, deltas, coastal wetlands and flooded lake margins (Fig. 1e). Areal coverage of the RFWs is by definition larger than the area of wetlands in all three input datasets (Fig. 1b-d), which were selected to be representative of different types of data acquisition (sensors and wavelengths). Therefore, they correspond to different definitions of inundated areas, and their contribution to the RFW map is fairly different. In particular, the shared fraction of the three input maps is minuscule (5% of the total RFW land surface area coverage), and is mostly composed of the large river corridors and ponds which are detectable by satellite visible range imaging techniques in the JRC dataset. The latter misses most understorey inundations, which are better identified by the ESA-CCI dataset owing to specific vegetation classification. Finally, owing to the use of microwave sensors, GIEMS-D15 extends over larger areas since it captures flooded areas and wet soils, below most vegetation canopies unless the densest ones (Prigent et al., 2007). Besides, the distribution of wetlands in GIEMS-D15 involves downscaling as a function of topography, and can be very different from the other datasets. Hence, 58% of RFWs are solely sourced from GIEMS-D15, mostly in the South-east Asian floodplains, North-east Indian wet plains and rice paddies, and in the Prairie Pothole Region (in Northern US and Canada). The ESA-CCI contribution is mainly found in the Ob River basin where wetland vegetation exists but wet soils are not easily detected by visible (JRC) or microwave (GIEMS-D15) observation. Due to its high resolution, JRC surface water adds small-scale wetlands such as patchy wetlands, small ponds and oases (0.4% of the land surface area).*"
- "*5.1 Uncertainty of the underlying layers (First paragraph)*
*It must be stressed that the uncertainty of the proposed CW maps is high, owing to several factors impeding the accuracy of the RFW and GDW maps. The uncertainty of the RFW map comes from the three input layers (ESA land cover, GIEMS-D15, and JRC surface water), and the lack of accuracy of the remote sensing products they rely on (shown by their large range of global flooded extents, from 1.5 to 7.7% excluding lakes). Of particular relevance is the uncertainty of GIEMS-D15, which contributes a lot to the high fraction of RFWs, and exhibits a small overlap with the other two datasets (less than 10% of inundated areas in GIEMS-D15 are confirmed by either ESA land cover or JRC surface water). Taking GLWD as a reference, Adam et al. (2010) concluded that inundation extents are overestimated in GIEMS (0.25° product of Papa et al., 2010) over parts of Northern Europe and India "because very wet soils may be wrongly identified as inundated", but this kind of error is not a major issue to identify wetlands, instead of inundated areas, as targeted by the CW maps. In India and South-East Asia, GIEMS-D15 also includes areas with flooded irrigation, including large rice-paddies, which correspond to artificial wetlands, not recognized in GLWD. Eventually, it is plausible that the RFW contribution from GIEMS-D15 is overestimated, but it must also be underlined that GLWD is not an exhaustive reference as it likely lacks some wetlands, as reported by Adam et al. (2010) and in section 4.2.*

**Comment 3:** "It would be helpful for the reader to draw rectangles corresponding to the sub-regional foci where the authors discuss some of the features of the datasets, e.g. in Fig. 6 (P13L20-21), Fig. 7 (P14L5-6), and in Fig. 10 (L15P22)."

**Reply:** These figures are corrected with rectangles showing regions of importance or disagreement. These figures are inserted at the end of this reply.

**Modification:**

- In figures 6, 7 and 10, rectangles are drawn around Negro river basin, Llanos de Moxos, upstream Mekong River and upstream White Nile River.

**Comment 4:** "The conclusion is quite long and sometimes feels like a discussion. In addition, it tends to reformulate the main results (e.g. P18L22-27) instead of providing conclusive remarks and outlook. Please consider avoiding too much repetition from Sect. 3, and passing elements of the conclusion over to the discussion."

**Reply:** We agree with the Referee#1 that some parts of the conclusion are almost repetitions of Sect.3 and we will revise the conclusion section in order to better represent the outlooks and concluding remarks. Many modifications are made. Some elements from the conclusion are moved to 5.1 (inserted to previous comments).Suggestions for summarizing the conclusion are explained below:

**Modifications:**

- In the conclusion, the first four paragraphs are summarized as follows: (in place of P17L36:P18L37)

*In an effort to develop a comprehensive global wetland description, we merged regularly flooded wetlands (RFWs) and groundwater-driven wetlands (GDWs) to develop composite wetland (CW) maps, under the assumption that both RFWs and GDWs are relevant although not exhaustive. The corresponding maps were produced globally at high resolution and two CW maps were selected based on comparisons with global and regional evaluation datasets. Their validity is particularly supported by the good match with the MPHFM dataset developed by Berthier et al. (2014) over France, because it was tailored to comprehensively include flooded and non-flooded wetlands with calibration against hydromorphic soils and validation against local surveys. With a total wetland fraction around 21% of the global land area, these CW maps are in the high-end of the literature, together with recent estimates also recognizing the contribution of groundwater-driven wetlands (Fan et al., 2013; Hu et al., 2017). It must be stressed that these high-end estimates, including ours, correspond to potential wetlands, as they neglect most wetland losses due to human activities, which may reach 30 to 50% of undisturbed or potential wetlands (Finlayson et al., 1999; Sterling and Ducharne, 2008; Hu et al., 2017). Overall, many uncertainties prevent from conclusively demonstrating that the CW maps are correct, in terms of patterns and extent, but it is also the case for any wetland mapping effort at the global scale that extends the definition of wetlands beyond inundated zones.*

- The rest of the conclusion remains similar with slight changes
- *Most of the* Second, third and fourth paragraph: (in the place of P18L6-L36): is modified and moved to *5.1 uncertainties of the underlying layers*
- Fifth paragraph: (in the place of P19L8-L13) is moved to : *5.1 uncertainties of the underlying layers*

- *"5. Discussion*

  *5.1 Uncertainty of the underlying layers"* (rest of the subsection)

…

[revised manuscript text omitted]

**SPECIFIC COMMENTS**

**P6, L34:** "Can the authors explain why they decided not to consider the permafrost effect for Ks estimates? Especially given the likely influence on the performance of TCTrI"

**Reply:** As shortly explained in the original manuscript, Gleeson et al. (2014) used a permafrost zonation index (PZI) from Gruber (2012) and defined all pixels with a PZI larger than a constant threshold as permafrost. They then assigned the hydraulic conductivity of all permafrost zones homogenously ($K_s = 10^{-13} \, m/s$). This ignores the diversity of different permafrost zones such as discontinuous and sporadic permafrost and instead considers all of them as continuous with constant active layer depths. Additionally, using this constant value leads to concentration of wetlands (derived from TCTrI) only over Siberian tundra and taiga forests, large parts of the Canadian Arctic Archipelago, Greenland and over well-known arid/semi-arid areas like the Kazakh uplands. In this way, many important wetland areas over equatorial and mid-latitudes like those in Amazon River basin, Congo basin and wet plains of Indian subcontinent are overlooked in these maps.

This is shortly pointed out in 2.4.4. We will modify 2.4.4 to better convey this message.

**Modification:** Underlined parts are the added phrases

- P6,L34: "*To consider the permafrost effect, Gleeson et al. (2014) used maps of the permafrost zonation index (PZI) from Gruber (2012) and homogenously assigned a rather low hydraulic conductivity ($K_s = 10^{-13}$ m/s) for areas with PZI>0.99. These areas with minimum hydraulic conductivity are limited to Siberian taiga forests and tundra, large parts of the Canadian Arctic Archipelago and Greenland. There is, as a result, a very large contrast in the hydraulic conductivity values between permafrost and non-permafrost zones leading to huge spatial discontinuities in the transmissivity map (considering constant depth for the permeable layer). For our calculations, we rasterized the vector polygons of $K_s$ without the permafrost effect to 15 arc-sec resolution.*"

**P10, L16:** "The transmissivity (Tr) map used here using GLHYMPS is at a much coarser resolution (1°) than the TI map (15 arc-sec). This mismatch is in addition likely to have a larger impact on dataset accuracy of TCTrI than the mismatch present in TCI ($P_e$ is at 0.5°), since intra-grid Transmissivity variability, from fine-scale heterogeneities, is likely higher than that of $P_e$. In a way, the lesser performance of TCTrI (or at least its little added value as compared to TCI) was thus somewhat predictable? "

**Reply:** This is a very good point raised by the Referee#1. But we first should correct that the resolution of the transmissivity maps is not constant in different parts of the world and as noted in 2.4.4, the GLHYMPS hydraulic conductivity map (base map for transmissivity) is formed of polygons of different sizes with an average of 100 km$^2$ (~0.1° at equator). So, the average resolution of the transmissivity map is higher than that of the climate forcings ($P_e$). The performance of the TCTrI is therefore not necessarily worse than TCI because of resolution. However, at zones where the spatial accuracy of the transmissivity map can be assumed too low by comparison to regional geological or soil maps for instance (Siberia, most of African continent, Indian peninsula, central Asia and upland Amazon basin), there is indeed a large mismatch between GDW-TCTrI maps and existing pattern of wetland distribution (Sect. 3.2.3).

**Modification:** There was no modification necessary for this comment.

**P13, L10:** ""covered above" is vague, can the authors specify which datasets they are mentioning? "

**Reply:** "This is right, and to make the sentence clearer, we revised the phrase giving an example of a study (Saunois et al., 2016) in which the overlap of GLWD (Lehner and Döll, 2004) and SWAMPS (Schroeder et al., 2015) is used to represent the wetland extent to calculate the methane emission from natural wetlands.

**Modification:** Underlined parts are the added phrases

- P13L5L10: *"The larger wetland fraction in MPHFM and CW maps is consistent with the work of Pison et al. (2018), who found that (wetland-driven) methane emissions over France deduced from atmospheric inversion were almost a third higher than those simulated by state-of-the-art climate models driven by global wetland datasets like the overlap of GLWD and SWAMPS used in Saunois et al. (2016)."*

**P13, L25:** "*in line with recent estimates of wetlands and peatlands*": references to published works would be needed here.

**Reply and modification:** The intended literature is the map of Hess et al. (2015). The reference will be mentioned.

**TECHNICAL COMMENTS**

**P6, L34:** The symbol or notation after "polygon of" is not properly rendered

**Reply and modification:** the symbol is $K_s$, which will be corrected in the new manuscript.

**P13, L4:** "Fig. 5j", instead of "Fig. 5"

**Reply and modification:** Fig. 5 is changed to Fig. 5i,j since it is pointing at both CW maps.

**P15, L17:** Instead of "CW-TCI(15%)" , "CW-TCI(6.6%)", "CW-TI(6%)", dropping the "%" in the name (CW-TCI15, CW-TCI6.6, etc.) could help avoiding confusion when these names are used to describe wetland extent.

**Reply and modifications:** It's a very good idea. We will modify the names of the generated maps in the text, tables and figures accordingly.

[Figure]

**Figure 6: Maps of the Amazon River basin wetlands according to different water and wetland datasets: (a, b, c) components of RFW, (d, e, f, g) evaluation datasets, (h, i, j) datasets generated in this study. The panels also give the mean areal wetland fraction of each dataset in the study area (using the mean fraction of each fractional wetland class of GLWD-3, cf. Sect. 2.5.1). The bounds of the basin are taken from Hess et al. (2015).**

[Figure]

**Figure 7: Maps of the South-East Asian wetlands according to different water and wetland datasets: (a, b, c) components of RFW, (d, e, f) evaluation datasets, (g, h, i) datasets generated in this study. The panels also give the mean areal wetland fraction of each dataset in the study area (using the mean fraction of each fractional wetland class of GLWD-3, cf. Sect. 2.5.1). The bounds of the study window are (5°-28°N, 82°30'-108°E).**

[Figure]

**Figure 10: Maps of the Sudd swamp wetlands according to different water and wetland datasets: (a, b, c) components of RFW, (d, e, f) evaluation datasets, (g, h, i) datasets generated in this study. The panels also give the mean areal wetland fraction of each dataset in the study area (using the mean fraction of each fractional wetland class of GLWD-3, cf. Sect. 2.5.1). The bounds of the study area are (4°30'-14°N, 24° 30'-34°E).**

[Figure]

**Figure R1: a) RFWs and b) zones where at least two of the inundation datasets agree over western Europe**

[Figure]

**Figure R2: a) RFWs and b) zones where at least two of the inundation datasets agree over Ob river basin**

---

## Author Comment (AC2) · 7 Dec 2018

We thank the Referee #2 for his/her comments and suggestions. They will help improve the manuscript. We will address comments in the order discussed by the Referee starting with "General comments". In the following Referee's comments are in black, replies in blue and modifications of the manuscript we propose in green. New phrases added to the text are underlined.

As a preface we would like to mention that most of the referee's comments were regarding the uncertainty or inaccuracy of the datasets that we used as input or validation base-maps for those we developed. In particular when we used GIEMS-D15 as an input dataset to develop the regularly flooded wetlands (RFW) map, we considered it to contain part of the true distribution of wetlands and a viable source. It is the same for Fan et al. (2013) and GLWD, here used as validation datasets. The basic assumption in this scientific effort was that each dataset contain part of the correct spatial distribution of wetlands but none is comprehensive. As a result we did not try to validate our composite wetland maps in a traditional way, but rather compare and evaluate them with other sources for wetlands.

In the referee's report one of the comment blocks (A) is divided into three parts within the text. We decided to discuss them together in one block

**GENERAL COMMENTS**

**Comment 1:** "The dataset is accessible via the given identifier and is documented. It is significant, unique, useful, and complete. It is unique and complete in that it combines RFW and GDW. It is useful as there is currently a low level of knowledge with respect of the location and size of wetlands at the global scale and there exists a demand for an improved knowledge. The dataset is usable in its current format and size.

The article itself is not yet appropriate to support the publication of the datasets.

A) In particular, the uncertainty of the data is not yet discussed enough (read more details below).

**Reply:** A new subsection is added to section 5 (discussion) to discuss the uncertainty of the data. Following the recommendation by Referee #1, it includes parts of the Conclusion which were a discussion of the maps and methods. It also includes new paragraphs in response to the detailed comments of Referee #2, as explained below.

**Comment 2:**

….

Regarding a clearer presentation of uncertainty, please discuss: 1) the uncertainties of the GIEMS-D15 (and in particular the underlying GIEMS dataset)

….

Regarding 1) the work of Adam et al. 2010 indicates that GIEMS overestimates inundation area in the Netherlands and Northern Germany and more so in India (if rice-paddies are taken into account), likely due to confounding wet soils with inundated areas. This was later supported by unpublished work (Master thesis of Matthias Konig). Here, the author compared Landsat ETM+ scenes with GIEMS and found a strong overestimation of inundation extent by GIEMS in Ganges-Brahmaputra and Parana river basins during and after months with more than 100 mm rainfall. Evaluating the presented RFW map for the Netherlands

and northern Germany (around Hamburg), I do not find it realistic that so many km$^2$ of land area inundated on average in one out of twelve months. I suggest providing the dataset on Google Earth such that the validity/uncertainty of the data set can be judged more easily based on local knowledge about e.g. the mean annual maximum extent of inundation that was predominantly used to produce RFW.

**Reply:** We agree with the referee that the uncertainties of the GIEMS datasets is not discussed. Our approach was based on the fact that the underlying wetland datasets used correctly contain parts of the wetland distribution. The scope of the work was not set to consider the validity of each input dataset, given they are already published.

For example, over the Netherlands (mentioned by referee#2), 21% of the country's land surface area is recognized as Ramsar wetland sites (more than 9000 km$^2$). These sites are the ones under international protection and it is not surprising to have 48% of the country's land surface covered by wetlands in RFW map, since many wetland sites are not recognized in Ramsar convention or are small in size. Moreover, based on maps of observed water table depth, the groundwater level is almost at the surface for most of the Netherlands Fig.R1a. This is in contrast with the wetlands in GLWD with only less than 4% coverage of the country. It appear as the RFW and CW-WTD (Fig.R1c,d) capture significantly better the pattern of shallow water table depths all over the Netherlands, in particular coastal wetlands from Rotterdam in the north to the Westerschelde estuary in the south.

[Figure]

**Fig.R1: (a) the observed water table depth in the Netherlands, taken from the Fan et al., (2013), supplementary material; (b) wetlands in GLWD; (c) RFW map; (d) CW-WTD. The percentages in each window show the ratio of wetlands' area to Netherlands's surface area.**

Due to extensive urbanization in western and northern Europe and intense agricultural activities in the Indian subcontinent, a lot of pristine wetlands were drained or transformed into irrigated/rainfed farmland. Although typical wetland vegetation and animal habitat have been destroyed in these zones, they can still count as natural potential wetlands where wetlands could have existed. More importantly, these areas often have wetland properties like high soil moisture most of the year, and may even be

flooded seasonally (e.g. rice paddies in north eastern India), which probably explains why they are seen in GIEMS-D15. This is also true for the floodplains of most rivers, even when regulated (e.g. Seine River and the surrounding flat lands). Additionally, these very wet soils are often zones of intersection between RFWs and GDWs where soil surface becomes flooded not due to overbank inundation but from excess groundwater discharge to soil column. Given our purpose to identify wetlands, we feel that errors of GIEMS because this dataset cofounds inundated areas with wetlands is not a major issue for our study.

We modified the manuscript by adding chapter 5.1 to further explain the uncertainty of input/validation layers

We also will provide the *.kmz for visualization in Google Earth for the final manuscript. It will be added to open access directories in PANGAEA data repository. Yet, since the size of raster datasets is large we put a lower resolution of the developed map as .kmz files.

**Modification:**

 *"5.1 Uncertainty of the CW maps and underlying layers* First paragraph

*It must be stressed that the uncertainty of the proposed CW maps is high, owing to several factors impeding the accuracy of the RFW and GDW maps. The uncertainty of the RFW map comes from the three input layers (ESA land cover, GIEMS-D15, and JRC surface water), and the lack of accuracy of the remote sensing products they rely on (shown by their large range of global flooded extents, from 1.5 to 7.7% excluding lakes). Of particular relevance is the uncertainty of GIEMS-D15, which contributes a lot to the high fraction of RFWs, and exhibits a small overlap with the other two datasets (less than 10% of inundated areas in GIEMS-D15 are confirmed by either ESA land cover or JRC surface water). Taking GLWD as a reference, Adam et al. (2010) concluded that inundation extents are overestimated in GIEMS (0.25° product of Papa et al., 2010) over parts of Northern Europe and India "because very wet soils may be wrongly identified as inundated", but this kind of error is not a major issue to identify wetlands, instead of inundated areas, as targeted by the CW maps. In India and South-East Asia, GIEMS-D15 also includes areas with flooded irrigation, including large rice-paddies, which correspond to artificial wetlands, not recognized in GLWD. Eventually, it is plausible that the RFW contribution from GIEMS-D15 is overestimated, but it must also be underlined that GLWD is not an exhaustive reference as it likely lacks some wetlands, as reported by Adam et al. (2010) and in section 4.2."*

**Comment 3:**

…and 2) The GDW dataset that, with the resulting 15% of global land area being identified as GDWs by global groundwater modelling (Fan et al. 2013), also constrains the topographic index-derived GDWs.
…
Regarding 2), I would think that with the steady state groundwater flow modelling approach of Fan et al. (2013), an overestimation of areas with shallow groundwater tables is very likely. In this approach, the location of surface water tables is not taken into account but groundwater flows out freely and is removed if the groundwater table exceeds the land surface elevation. However, rivers are in most cases incised into the land surface, i.e. the river channel forms a long depression as compared to the surrounding floodplain such that the groundwater table will be lowered in comparison to the Fan et al. (2013) approach because the drainage level is lower.

**Reply:** The uncertainty of the wetland thresholding (here set to 15% following the wetland fractions in Fan et al., 2013), is discussed in brief by choosing two approaches for defining the thresholds as discussed in 3.2.2 under "Two index thresholds for two global GDW fractions". It should be restated that as said in the article, in absence of a globally agreed upon wetland definition the threshold choices remain always subjective. But by testing a set of thresholds (both on wetland fraction and WTD), we were capable of evaluating the sensitivity of wetland extents to this important element (Fig S3 and Table S1 of the supplementary)

We agree with the reviewer that groundwater depths might be underestimated (shallower water table depths) because of the simplification in modeling exchanges with streams. This is more obvious in areas where river incisions are deep into aquifer. Yet this uncertainty is very difficult to quantify and is beyond the scope of this work. It should be also noted that most wetland hotspots of the world are located in areas with very flat topography and slow moving rivers (e.g. Hudson Bay lowlands, Ob river basin, Gang-Brahmaputra delta), where there is not a large difference between river water levels and surrounding lands. The groundwater near rivers is lowered to a much greater extent in hillslope but in permeable lowlands groundwater level is very close to water level in rivers.

**Modification:**

*5.1 Uncertainty of the underlying layers*

*…*

*Regarding the GDW maps, two major sources of uncertainty can be identified, related to modelling and thresholding. Whatever the involved GW modelling (simplified based on wetness indices, or direct like in Fan et al., 2013), a major challenge is to define thresholds on TI or WTD to separate the wet and non-wet pixels. Following the existing literature, we defined wetlands as areas where the mean WTD is less than 20 cm, and this WTD threshold was translated into the TI threshold defining the same global wetland extent (15%). Any error on this extent because of modelling errors will propagate to TI-based wetland mapping. In particular, the steady state assumption and 1-km resolution used by Fan et al. (2013), as well as their imperfect input data, only leads to a "first-order estimate of global land area likely affected by shallow groundwater", according to the authors. Nevertheless, the threshold choices remain subjective in the absence of consensual global wetland map and definition, and the related uncertainty in wetland extent was shown to amount to a few percent of the total land area based on sensitivity analyses for reasonable values of the different thresholds (supplementary section S2, Fig. S3 and S4).*

**Comment 4:**

B)    "The authors should explain more in detail how this dataset could be used for what purpose, dependent on the (large uncertainties). Particularly difficult to utilize is the information on GDW. How could, for example, the estimated GWD distributions that are based on simple steady-state modelling of groundwater tables, without interaction with surface water bodies and without taking into account human activities, be actually used in large-scale land surface modelling? Maybe focus on results for France, e.g. Landes."

**Reply:** This dataset is representative of the maximum extent of wetlands (both RFW and GDW) without considering the impact of human activities. Therefore it can be used in large scale land surface modelling

as a boundary condition for wetland modellings. The results of the steady state modelling is interesting as it bounds the wet fraction of each cell to a static value.

The following underlined sentences are added to section 6 to better explain the potential applications of the proposed maps.

**Modification:**

- *6 Data availability and application*

*"These classified maps are believed to be useful for hydrological or land surface modelling by assigning specific properties or processes to the places identified as wetlands or floodplains. The RFW maps can used in global hydraulic models, for instance to constrain the buffering capacity of floodplain reservoirs, recently identified as critical parameter for peak discharge simulation (Zhao et al., 2017). More originally, the CWs can be viewed as the spatial support of a particular "hydrotope" (Gurtz et al., 1999; Hattermann et al., 2004), i.e., the hydrological analogue of plant functional types (PFTs) for vegetation properties and processes (Lafont et al., 2012). In these hydrotopes, the extent of which can be deduced from the CW maps, specific models can be used to quantify methane production or denitrification by wetlands, for instance, especially if combined with dynamic modelling of the saturation degree within the wetland fractions (Hesse et al., 2008; Post et al., 2008). Depending on the particular purpose, the user can choose to define a lumped hydrotope merging RFWs and GDWs, thus corresponding to the CWs; or to separate RFWs from non-regularly flooded GDWs, the latter being mapped by excluding RFWs from CWs. As an example, the CW-WTD map was recently used to calibrate a cost-efficient TOPMODEL approach aiming at simulating the dynamics of peatland area and related carbon fluxes (Qiu et al., 2018). Although the CWs do not necessarily match areas with specific wetland vegetation, they can also be used to locate areas deserving specific PFTs, corresponding to plant species adapted to low water stress or shallow water table (e.g. Fan et al., 2017). Another promising application is to constrain GW modelling in land surface models, by locating the areas where GW are sufficiently shallow to influence soil moisture by capillary rise, as done by Vergnes et al. (2014) based on arbitrary topographical considerations. Finally, provided the CWs maps offer a sufficiently accurate description of potential wetlands, they can be combined to maps of land cover change to better quantify wetland losses, and the related impact on the global water or biogeochemical cycles (e.g. Sterling et al. 2013)."*

**Comment 5:**

  C) "Please state very clearly that the wetlands identified in this dataset do not correspond to wetlands with typical wetland vegetation"

**Reply:** The definition of wetlands is explained in the introduction of the paper in the fifth paragraph like the following: "*This approach leads to a definition of wetlands as areas that are persistently saturated or near saturated because they are regularly subject to inundation or shallow water tables*". In this way, wetland definition is independent of the vegetation in these areas. However, to make it completely clear we will add a sentence to explicitly mention the independence of wetlands in this article from typical wetland vegetation (explained below)

**Modification:**

- Introduction 5th paragraph

*This approach leads to a definition of wetlands as areas that are persistently saturated or near saturated because they are regularly subject to inundation or shallow water tables. This definition is focused on hydrological functioning, and is not restricted to areas with typical wetland vegetation. In this context, although inundated areas and zones with shallow groundwater partially overlap and share similar environmental properties, they cannot be detected using a single method.*

**Comment 6:**

D)      "Validation of the dataset is not done correctly. I think it is not appropriate to use GDW-WTD derived from Fan et al. (2013) as validation dataset as this dataset is the basis of your estimates of GDW in all CW maps, directly or indirectly. It is not correct to say that the other CW maps (those were topo indices were used) are independent from GDW-WTD as the total area of GDW (15% of land area) that is only distributed via the topo indices is prescribed by GDW-WTD. Do not use GDW-WTD as a validation dataset"

**Reply:**

We understand the referee's concern about using GDW-WTD (simulation results of Fan et al., 2013) both as input and validation datasets. But some points are worth mentioning in reply to this comment:

- What we have done in the validation part of the article is not a traditional validation procedure. As clear from the comparison between Hess et al. (2015) and MPHFM (based on national studies) on the one hand, and the other validation datasets (global) on the other hand, actual wetland extents can be very different in these two kinds of datasets. That is why we want to emphasize that a traditional validation procedure where the developed map replicates the validation dataset was not the objective of our study. Indeed we claim the wetland maps we are presenting are detecting some wetlands that have not been by previous studies. We agree that the CW maps derived from index thresholding is partially dependent to Fan et al. (2013)'s simulation result because of the prescribed wetland fraction. But the spatial pattern of wetlands in these CW maps is completely independent, due to completely different methodology.
- On the other hand, for global comparisons with CW maps, there are simply no other available wetland maps to our knowledge.
- Although it is always subjective to claim one dataset is over/underestimating the wetland extents, we can assume that each of the validation datasets GLWD-3, Hu et al. (2017) and GDW-WTD lack accuracy in different places of the Earth and using them altogether in a validation processes can improve the overall validation quality.

Based on this discussion, we believe that keeping the GDW-WTD as a validation dataset brings added value to the study.

**Modification:** underlined sentences have been added to section 4.1

*"4.1 Spatial similarity assessment*

*A difficulty inherent in the validation of any wetland map is the vast disagreements among available datasets and estimates. In this paper, we used independent validation datasets (explained in Sect. 2.5) that are not used in any step as input to our final products, but we made an exception for the GDW-WTD (derived from Fan et al., 2013), although it is a direct input to CW-WTD, and we used the total wetland fraction of GDW-WTD (corresponding to WTD ≤ 20 cm) to define the TI thresholds behind the TI-based CW maps. This exception is considered for two reasons. Firstly, we focus here on spatial patterns, which are completely independent between TI-based CW maps and GDW-WTD, because of very different GW modelling assumptions and input data. Secondly, we also focus on wetlands rather than inundated areas, and on their detection under dense vegetation: GDW-WTD is one of the very few global datasets with these properties, but it results from a different method than Hu et al. (2017) and GLWD-3, so it can help enriching the uncertainty discussion. All seven developed CW maps and the RFW map were evaluated using the spatial coincidence, Jaccard index and spatial Pearson correlation coefficient with respect to the validation datasets over the globe and in several regions, the latter of which are discussed in detail below."*

**Comment 7:** "It would be interesting to better understand the map of potentially wet zones in France. With 23% of France being wetland, or rather a potentially wet zone, is this based on having 23% of France covered by soils with hydromorphic features?

**Reply:** The French potential wet zones map is actually based on a combination of 1) a map derived from topographic index with thresholds coming from soil maps (only available in parts of the French territory) classifying hydromorphic soils; 2) a map of area with small elevation difference with streams using HAND algorithm (Height Above Nearest Drain). In areas with high infiltration potential it is claimed that topographic index is not appropriate and elevation difference to streams is used instead. In this study, the extent of areas that we can assume to be related to hydromorphic features is almost 18% of the France metropolitan area. The remaining 5% of wetlands in MPHFM are composed of pixels with equal or lower elevations to streams.

**Modification:** 2.5.4 Modelled potentially wet zones of France (in place of P8L2)

*"…The wet fraction defining the threshold in each hydro-ecoregion is the fraction of hydromorphic soils (extrapolated from local soil maps to almost 18% of France metropolitan area) taken from national soil maps at 1:250,000 (InfoSol, 2013)."*

"Do you have an idea about the actually wet area in France, taking into account human impact?"

**Reply:** There is not another map of wetland areas in France to our knowledge. As stated in the paper the scope of this work is not to consider the actual human impact on wetlands globally.

"Are there maps for France for RFW?"

**Reply:** There are no national maps for this purpose and extracts from global maps have often been used in previous studies.

"Commissioning error in Berthier et al. (2014) is high (75% of validation points are wrongly classified), what does this mean for your comparison?"

**Reply:** These rather large errors in validating the model with point pedological data can be attributed to:

- The errors hidden in not taking into account all the parameters of forming wetlands
- Wetlands that have lost their wet character due to draining, agriculture, etc.
- Heterogeneity of the distribution of validation data used in Berthier et al. (2014)

As a result, although these errors could have been caused by inability of indices in predicting the location of wetlands, they also could have been driven from the impact of human activities on wetlands or uneven spatial distribution of validation point data.

**SPECIFIC COMMENTS**

**P4, L29:** "Why did you exclude only natural lakes and not also reservoirs that can be estimated in the GRanD database to cover 305,000 km$^2$? Please mention"

**Reply:** We did exclude reservoirs from the final wetland maps. This is done using the classification in HydroLAKES (itself using the GRanD database for reservoirs). We will explicitly mention that under 2.2 for clearer description.

**Modification:**

- 2.2 Lakes

*"…1.4 million individual polygons for lakes with a surface area of at least 10 ha, covering 1.8% of the land surface area. It also classifies artificial dam reservoirs which amount to 300 10$^3$ km$^2$ (Messager et al., 2016). The lakes' extent in HydroLAKES is …"*

**P6, L6:** "Modelled WTD was not constrained by observations by Fan et al. (2013) but only compared"

**Reply:** We agree with the referee. The mistake is corrected in the new manuscript.

**Modification:**

- *"…The modelled WTD was compared to observations available to the authors (more than one million observations with 80% of them located in North America)"*

**P7, L9-10 and Table 1:** "GLWD-3 values listed in Table 1 actually do not include lakes and reservoirs, not the Caspian sea (see Table 3 in Lehner and Doll 2004). Why do the values in the table differ from the values on page 7?"

**Reply:** We thank the reviewer for noticing this point. We will correct the footnote for table 1 in our manuscript. However we masked out all the lakes in HydroLAKES from all validation datasets including GLWD. Therefore, the lakes that were not considered in GLWD are also excluded from this map in values mentioned in Page 7. The extent of lakes not considered in GLWD is the reason for this difference.

**Modification:**

- Table 1. Footnote:

*"‡ Excluding Antarctica, glaciated Greenland, including lakes and Caspian Sea. Additionally the range in GLWD is different based on interpretation of fractional wetlands."*

**All manuscript:** "please explain clearly in the manuscript why the 15% assumption works better than the 6% assumption, by referring more strongly to Table S1."

**Reply:** In the manuscript under 4.1 it is explained why we prefer the 15% assumption over the 6% for CW-TCI15 and CW-TCI6.6. We will expand this discussion with regard to Table S1 both in manuscript and in the supplementary.

**Modification:**

- 4.1.1 Global analysis

*"…As demonstrated in Fig. 4a (and also Table S1), increasing the GDW contribution from CW-TCI6.6 to CW-TCI(15%), as an example, also improves the similarity criteria (except the SC for GLWD-3 and GDW-WTD), justifying the need to account for the GDWs to supply a comprehensive description of wetlands. This is clearer for the global spatial correlation values which all increase when the contribution of GDW is increased from 6.6% to 15% (Table S1: first row block)."*

- 4.1.5 Hudson Bay lowlands

*"…Surprisingly, GLWD-3 has a pattern notably similar to those of the other two validation maps due to the comprehensive wetland maps in Canada available to its developers. Moreover, HBL is one of the few regions where similarity indices sharply increase with increased GDW contribution (Table S1). The Jaccard index rises from 46 to 53 when increasing the total GDW extent from 6.6 to 15% between CW-TCI6.6 and CW-TCI15."*

- 4.1.6 Ob river basin

*"…However, CW-TCI15 appears to miss wetlands south of 60° N that are extended to the upstream Ob river basin near 50° N in both GDW-WTD and Hu et al. (2017), most likely due to the permafrost effect on wetland formation. With respect to the evaluation criteria, CW-WTD often displays better performances, although CW-TCI15 show the highest SPC. We also find that CW-TCI15 outperforms CW-TCI6.6 for all criteria/validation dataset combinations (Table S1)."*

**TECHNICAL COMMENTS**

**P6, L16:** "1-meter DEM?"

**Reply:** here by 1-meter DEM we meant 1-meter resolution.

**Modification:** DEM is replaced by word resolution

**P6, L26:** "illegible"

**Reply:** Corrected

**Modification:** $K_s$ was badly rendered in the text which is added